# The Liminal Space of Medieval Dance Practices: The Case of St. Eluned's Feast Day

**Laura Hellsten**

Systematic Theology, Åbo Akademi University, 20100 Turku, Finland; laura.hellsten@abo.fi

**Abstract:** This article scrutinizes the use of liminality as a term to understand medieval dance practices. With the case of the feast day of St. Eluned described in Gerald of Wales Itinerarium Cambriae, I first present common ways that historians and theologians have used the term liminality in order to describe historical depictions of feasts of saints where more unruly forms of movement and dancing have happened. I then analyze this specific depiction by Gerald of Wales through a combination of a kinesic approach and a hermeneutics of suspicion and charity. This approach shows that earlier understandings of dancing always being a problematic element in traditions of Christianity in the west needs to be nuanced. After this, I turn to the critique that Caroline Bynum Walker has brought up, concerning the use of the term liminality in the medieval context. Taking her critique seriously, I return to the story of St. Eluned by focusing on the lived religion from the perspective of the female characters in the setting. Finally, I also bring in Vincent Lloyd's distinction between rituals and liturgy, to further strengthen how theological discussions can bring in more nuanced and important additions in how we may understand chaotic forms of medieval dance in new ways.

**Keywords:** dance; theology; dance history; Wales; saints; liminality; relics; women; liturgy; ritual; hermeneutics of suspicion and charity; kinesic analysis

## 1. Introduction

In this article, I argue that dance creates theology, rather than being a mere reflection of it. I advocate for the need and benefit of mixed method approaches for both researchers in theology and dance history, when it comes to understanding the religious meaning and purpose of dance in historical records. With the specific case study of the celebration of St. Eluned's feast day found in Gerald of Wales's *Itinerarium Cambriae* I show how kinesic understanding, which is an approach used in dance historical studies, may enrich theological reflection. I further show how analysis of textual sources describing dance benefit greatly from both on-site visits and engagement with iconography. I also show how dance history may gain important insights from following theological arguments concerning the relevance of art as theology (Sarah Coakley's *Théologie totale*), as well as specific distinctions made in political theology between the concepts of ritual and liturgy.[1] Furthermore, I argue that dance historians need to become more nuanced in how they read and use texts by church authorities.

The main material of this article is a story found in Gerald of Wales's *Itinerarium Cambriae*. It is said to be the most detailed medieval description of traditional circle dances.[2] In most sources, when the feast of St. Eluned is brought up as an example of dancing in the early medieval period, only parts of the account are quoted.[3] Dom Louis Gougaud, who is often referred to in theological treaties on dance, sweeps over the account without much regard, simply grouping it together with the Kölbigk dancers as involuntary spasmodic movements.[4] He further remarks that the only purpose of telling stories like these was that they could be used as examples that warn people away from dancing.[5] Similar to the recent

writings on the Kölbigk dancers in *The Cursed Carolers in Context* (Renberg and Phillis 2021), I argue that there is much more than a warning to be found in this story.[6] Theologically, I argue that the story of St. Eluned's feast day should be read not primarily as an unruly form of dance, instead, it should be understood as a celebration of a feast day of a saint. For such an understanding, Peter Brown's use of the term liminality (based on Victor Turner's development of this concept) is a useful tool. However, I argue that for the terminology of liminality to function in this context it needs to be paired with other concepts brought forth by Brown, such as *reverentia, praesentia* and *potentia*. In this combination, the idea of liminality may open up for a theological view on how Christian communities understand their interactions with the divine as, potentially, politically and socially transformative. I further argue that reading dance kinesically, instead of through pre-set ideas about the Church always condemning dance, will help us understand the multi-layered potency of dance in the theological imagination of the medieval period.

In the second half of this article, I then move even further than this. I argue, along with Caroline Walker Bynum, that using the term liminality in a context containing examples of women and laity dancing, as well as in understanding the descriptions of the life of a female saint, the idea of liminality reaches its limits. Instead, I suggest a *hermeneutics of suspicion* and *charity*, which reaches for materials beyond the accounts given by male authors. This methodological approach opens up for reading the accounts created by male authors, both with a critical tone, and to find other venues through which one can engage creatively with the lived experiences of women doing theology. It is in this context that the difference brought forth by Vincent Lloyd between dance as a ritual, or dance as part of liturgy, will become significant. When dance is understood as a ritual it will, even within the context of the idea of liminality, mainly be read as a sign of a praxis of social cohesion. When, however, dancing is part of the liturgical expression of the community, it carries the capacity to even alter social norms. In this last section, I thus venture into understanding the account of St. Eluned's feast day in the context of dance creating theology.

In the end, this article thus offers two different ways in which to read this ethnographic depiction of dancing given to us by Gerald of Wales. It will be up to the reader to judge which of these bests fit their understanding of dance in the medieval period.

## 2. Background and Methodology

In articles dealing with the dance practices that can be found from the medieval period, liminality has been used as a framework to understand these events. Many of these cases refer to Victor Turner's theory of liminality, which I will return to in the next section. Examples of the medieval dancing discussed range from unruly carolers in Christmas stories to Gerald of Wales's explication of the dancers at the feast of St. Eluned.[7] In this article, I point out some of the possibilities of understanding dance with the help of liminality. Simultaneously, I argue that when one wants to understand medieval dance happening in and around churches, historians and theologians need to engage in a methodology that employs multiple frameworks of interpretation. Furthermore, both Peter Brown and Caroline Walker Bynum have highlighted the challenges that arise when historical research wants to engage with the lived religion of a specific social class and/or gendered community.[8] In this work, I will relate and connect those remarks to the praxis of dancing, arguing that understanding dance features at a saints day celebration requires more theoretical reference points than the framework of liminality may offer.

Historical studies interested in dance before the modern period run into a complex set of challenges when it comes to an understanding of the role and function of the dancing.[9] This dilemma is two-fold. On one hand, the fleeting nature of dance—leaving very few material traces behind—always presents challenges for textually focused academic research. Even in a time of video and film, the ambience of the surroundings and the sensory experience of moving in space cannot be fully captured or reproduced.[10] The *Medieval Folklore* encyclopedia explains that information about dance must be gathered from iconographic sources, literary references and sometimes musical evidence.[11] This is well exem-

plified in Kathryn Dickason's *Ringleaders of Redemption: How Medieval Dance Became Sacred* (Dickason 2020), where she elaborates with a comprehensive set of images and textual sources. However, as Dickason points out, there has been a strong propensity towards binary thinking when describing and interpreting medieval dancing.[12] Furthermore, in *Through the Bone and Marrow: Re-examining Theological Encounters with Dance in Medieval Europe* (Hellsten 2021). I explain that, particularly when it comes to a theological understanding of dance, the social imaginary of the researchers has prevented them from engaging seriously with dance as a source of theological knowledge. These are some of the reasons why this article adopts a *hermeneutics of suspicion* and *charity* as its methods of enquiry.

Sarah Coakley describes that her *Théologie totale*, with its *hermeneutic of suspicion* and *charity* is developed as a method which addresses the particular challenges that are brought forth when praxis meets theory, and previous generations of scholars have ignored more subversive aspects of theology. She explains that, especially when the patriarchal suppression of women, and questions of race and class are brought forth in the materials, a *hermeneutics of suspicion* is needed. Coakley's *hermeneutics of suspicion* is described as "an attitude towards texts that is sceptical about their surface appearance and seeks to reveal authors' unstated and in particular improper motives".[13] In the current article, such suspicion is applied to the primary text of Gerald of Wales, and the secondary textual interpretations that previous scholars have used when elaborating on the concept of liminality in their readings of Gerald of Wales.

Coakley points out that when researchers are caught up in hegemonic discourses and/or epistemic frameworks—over-emphasizing textual materials and language-based understanding of dance, for example—*suspicion* is one, but not the only way forward. Coakley, elaborating on Paul Ricoeur's writing, argues that artistic works can be used to unsettle previous politically and socially "dominant" ways of thinking.[14] The arts in general and in this article—dancing in particular—are seen not as an illustration of doctrine or a mere outcome of worship or cultural traditions.[15] Instead, dancing creates theology, enabling new expressions of doctrine while simultaneously articulating the lived religion of the people taking part in the dancing. This is something I wish to demonstrate when employing a *hermeneutics of suspicion* to my reading of Gerald of Wales's depictions of the feast day of St. Eluned.

Coakley further states that over-emphasizing *suspicion* also has its flaws. She is particularly wary of the kind of arguments that some strands of feminist theology bring forth, where it is presumed that skepticism will always have the last word. In these depictions, a position of powerlessness is more often reproduced than over-turned, when historical evidence could show us a contrasting story.[16] Coakley, along with Lisa Felski, thus brings forth the complement of a *hermeneutics of charity* to be kept hand in hand with our *suspicion*. The interpretative patterns suggested by Caroline Walker Bynum and Jane Cartwright, referred to in the second part of the article, want to contradict the tendencies of medieval scholarship to reduce the agency of women. In a similar manner, I aim, in the last part of this article, to present a reading of Gerald of Wales's story that centers on the female experiences of dancing together at the feast of St. Eluned. In her *Limits of Critique* (Felski 2015), Felski writes that:

> . . . both art and politics are also a matter of connecting, composing, creating, coproducing, inventing, imagining, making possible: that neither is reducible to the piercing but one-eyed gaze of critique.[17]

In order to understand how art, dancing and theological expressions from the medieval period may have focused on connecting, creating, co-producing and imagining new ways of being together, a researcher also needs to apply a gaze that is full of *charity* towards the materials that one encounters. Only by reading and interpreting historical materials with a sense of awe and wonder, will certain elements of the descriptions and depictions open up and reveal themselves to the researcher.[18]

In particular, when it comes to questions of religion and theological understanding of previous periods, earlier scholarship has presumed certain strands of thought to be superstition or irrational—wanting to psychologically, or even medically, explain away religious experiences.[19] In this article, I apply *charity* as a tool for perceiving the dancing and the religious experiences from the point of view of the practitioners and participants in the rituals. Thus, I frequently apply terminology used by Peter Brown, going beyond liminality, to depict the experiences of relating to martyrs and relics in the early church and early medieval period.[20] This includes, but is not limited to, terms like *potentia* and *praesentia* of the saint.[21] To contextualize the story of the feast day of St. Eluned liminality only takes us so far—deeply theological concepts are also needed. Further, in my attempt to combine a *hermeneutic of suspicion* and *charity*, I bring in the theological distinctions made between rituals and liturgies by Vincent Lloyd, in the later part of the article.

Finally, this brings us to this article's second and last aspect of methodological approaches. When looking at historical depictions of dancing, the most challenging concern is often, not the absence of primary dance sources, but rather, a challenging feature is the fact that texts cannot convey how movements functioned in space and time. As I have already established, Coakley's *hermeneutics of suspicion* and *charity*, opens up to include any art as theology, so it is willing to move beyond texts. However, the lack in her work is that she provides no tools for how to do this.[22] This is why I have turned instead to the recently well-argued points about historic dance depictions found in *The Cursed Carolers in Context* (Renberg and Phillis 2021).[23] In this book on medieval dance depictions, a whole chapter is devoted to the need for a kinesic approach to historical dance materials. Rebecca Straple-Sovers explains that kinesic analysis includes following the characters' gestures, manners and postures in their depictions.[24] It "foregrounds the bodily movements performed by characters within literary works and considers how they function as a system of expression and meaning-making".[25] This makes it possible to ask questions, such as how the dancers' movements supplement, contradict, or complicate their verbal interactions or the narrative content? What might be revealed about the characters, their interactions, states of being and emotional states, from the movements? To this, I would also add: What could be perceived about the meaning-making practices of being in relationship to the divine, and experiences of the closeness or presence of God, in the situation described by their movement patterns?

While doing a kinesic analysis of the movements in the story of Gerald of Wales—which I attempt to do in the first part of this article—is an essential addition to scholarship dealing with dance, it also has its limits. As Rebecca Straple-Sovers and many others have pointed out, experiences of emotions, sensations and the worldviews that one encounters in medieval sources were most probably completely different to what we may experience in our time and age.[26] Thus, applying a full archaeology of the senses approach to the story of dancing at St. Eluned's feast day is beyond the current project. Robin Skeates and Jo Day write in *The Routledge Handbook of Sensory Archaeology* (2020) about the need to use and explore methods such as critique, reflexivity, incorporation, ethnographic insights and analogies, direct experience, experimentation and reconstruction, imagination and artistic creativity, evocation and empathy when approaching historical materials.[27] A satisfactory account of the event incorporating these aspects would require much more space than what this article may offer. However, in gathering materials for this study, I did visit the places where St. Eluned is said to have passed through before her death. Thus, the latter part of this article will also include some kinesic analysis of the actual sites of pilgrimage, and my own movements in this particular space. In this manner, this article, with its *hermeneutics of suspicion* and *charity* and the kinesic analysis of the story of St. Eluned, introduces a new kind of multi-method exploration, while its main focus is on the critique of the use of the term liminality in previous research dealing with this specific story.

### 3. Dance and Theories of Liminality

The term liminal was first coined by Arnold Van Gennep, a folklorist writing about *Rites de Passage*, 1909 (Van Gennep 2013). The main idea of this term is that, in ritual passages from one state of being to another, there might be an in-between stage characterized by chaos or even destruction. At this threshold, behaviors, practices and modes of being under "normal" circumstances, that would not be accepted or perceived as distressful, are now given space as a path of transition or change.[28] In many of dance historian Gregor Rohmann's works, on exploring potential Christian modes of dancing in the early church and medieval period, this mode of relating to liminality plays a big role in his interpretations.[29]

After Van Gennep, the concept was picked up by anthropologist Victor Turner when, in 1967, he included an essay, "Betwixt and Between: The Liminal Period in Rites of Passage", in his book *The Forest of Symbols*.[30] In short, the theoretical discussion about liminality is formed around questions of both time and space. The scale of usage can also vary from describing an individual's life to small community groups, or sometimes encompassing whole societies and eras. This means that the term liminal has been widely used in describing moments, everything from abrupt events like death, illness and divorce, to specific rituals of passage. It has also been used to express longer passages of time, for example, periods of distress like war, famine or plague.[31] Furthermore, the term has been used for a specific "type" of people in a community, like a holy person, non-able-bodied individuals or holy fools, like St. Francis.[32] Finally, the concept has gained a stronghold in describing specific places and spaces. These may vary from borders of different types, holy wells, trees, or parts of a landscape, to areas like a graveyard or crossroads between life and death, and health and sickness, as the space of a relic or sauna may form.[33]

Theologically speaking, Peter Brown, in his *The Cult of the Saints—Its Rise and Function in Latin Christianity* (Brown 1981), is one of those authors who used Victor Turner's ideas about liminality to describe what happened at the shrines of the saints, as well as through the pilgrimage processions to graveyards or other holy sites, in the early church period.[34] He writes:

> Christians who trooped out, on ever more frequent and clearly defined occasions as the fourth century progressed, experienced in a mercifully untaxing form the thrill of passing an invisible frontier: they left a world of highly explicit structures for a "liminal" state.[35]

The walking out of the rigid structures of a city landscape and ordered society opened the way for entering a space where a new kind of community could be built, and where behavior that otherwise could be characterized as unruly, or inappropriate, could be unleashed.[36] In Brown's understanding, the shrines of saints were not just places where exorcism and other potentially unsettling practices could occur. They were also spaces where the *potentia* and *praesentia* of the divine—mediated by the relic—could transform the social status of individuals and the norms prevalent in society.[37]

Not only Gregor Rohmann, but also Nancy Mandeville Caciola has used at least part of this terminology to understand the dancing of the medieval period. They have looked at the dances that took place at the pilgrimage sites of St. Vitus and St. John, the famous "Dance Mania" starting in Kölbick around 1017/21, and the account of frenzied dancing described by Gerald of Wales in *Itinerarium Cambriae*.[38] The implicit starting point in these authors' writings seems to be that dancing is somehow problematic and needs to be explained, and that a dancing saint is out of the norm.[39] Liminality is then used to describe how we can make sense of the dancing found in these events. Both Rohmann and Caciola refer mainly to the space of the graveyard and the body (as an ambiguous entity) as a liminal place where dancing could happen.[40]

Very few dance scholars or medieval historians use Brown's terminology of *potentia* and *praesentia* to explain how we might understand the importance of this kind of dancing

to the community, or as a phenomenon with a theological significance. The exception to this rule is Caroline Walker Bynum. In *Wonderful Blood—Theology and Practice in Late Medieval Northern Germany and Beyond* (Bynum 2007), she brings forth the idea that ambivalent practices—a category in which I include more chaotic forms of dancing—could also have played a part in the activation of the relic.[41] In her depictions, just like those brought forth by Brown and Caciola, the relics did not facilitate miracles on a continuous basis. Instead, the birth, death and translation days of relics, in combination with an interactive component of the gathered community, were key in creating a situation where the *potentia* of the saint became visible to the community.[42] In combining Peter Brown's and Caroline Walker Bynum's descriptions of what happened at pilgrimage sites and shrines of saints, dancing could play a part in activating a relic and displaying a saint's *praesentia* at the site.[43] Finally, dancing in combination with healing could further be understood as a sign of the *potential* of the saint.[44] To exemplify, I turn to the description of the feast of St. Eluned.

## 4. Gerald of Wales and the Itinerarium Cambriae

The *Itinerarium Cambriae* is described as the eyewitness account by Gerald of Wales (1146–1223), derived from his tour through Wales.[45] It can be read as somewhat akin to a diary of an ethnographical expedition.[46] Carl S. Watkins, in *History and the Supernatural in Medieval England* (Watkins 2007), claims that Gerald of Wales was often unsympathetic to the Welsh culture, which he frequently portrayed as uncouth, barbarous and alien.[47] Geraldine Heng further explains that Gerald of Wales was particularly ruthless towards the Irish. Describing them, based on their pastoral living and economic practices, as uncivilized, backward and lacking in morals. With the lack of any physiological or even religious differences between the Irish and the English, Gerald of Wales took to the depictions of customs, cultural norms and manners to depict why the English needed to take initiative in this part of the world. He further positioned himself as a leader of the project of bringing the Irish towards evolutionary improvement and instruction.[48]

In another of his works, the *exempla* or *Gemma Ecclesiastica*, Gerald of Wales focused on giving instructions to the clergy.[49] Watkins argues that Gerald's accounts were clearly written with an eye on Rome, and a hope to curry favor in his battle for the bishopric of St. Davids.[50] In the *Gemma Ecclesiastica*, Gerald devotes a whole chapter to speaking against the dancing and singing of vulgar songs in churches and cemeteries.[51] He refers to the Councils of Toledo, which stated that dancing and singing during feast days of the saints should be abandoned for the sake of participation in the divine office. He further declares that it is the task of the priests and judges to make sure this is implemented.[52] He then continues by presumably quoting St Augustine, who stated that the interior part of the church, the oratory, was named so, because it was for the purpose of prayer (*ora = prayer*), nothing else.[53] Based on these accounts, it seems that Gerald of Wales was a person who, if he found problematic aspects of lived Christian faith amongst the people he met, was bound to report them. He also seemed more than willing to explain what he, as a guardian of the Church, had to offer in correcting such behavior.

It is due to these depictions that it is interesting to see that when he turned to the *Itinerarium Cambriae* and its description of the feast of St. Eluned, his account lacks a condemning attitude towards singing and dancing.[54] Even though the description is one of rural practices and dancing in the cemetery, his concerns about problematic behavior from the laity during the feast day of the saint, are missing. Something else was at stake in this story. Let us now turn to an in-depth reading of Gerald of Wales's writing about the celebrations of St. Eluned's feast day, from what may have been 1 August, around the year 1190.[55]

In *Butler's Lives of the Saints* (Butler 1981), Herbert J. Thurston and Donald Attwater write that Gerald of Wales was an archdeacon of the city of Brecon from 1175. Furthermore, he lived for over 20 years in Llanddew, only a few miles away from the shrine of St. Eluned, making both the region and the feast a space and place he would potentially have visited

several times.[56] In Gerald's account, he first explains the scenery of where the feast of St. Eluned is placed.

Below is a picture of the space outside of Brecon (which I visited 24 September 2019) in Slwch Tump, which is described as the setting of St. Eluned's shrine (see Figure 1).[57]

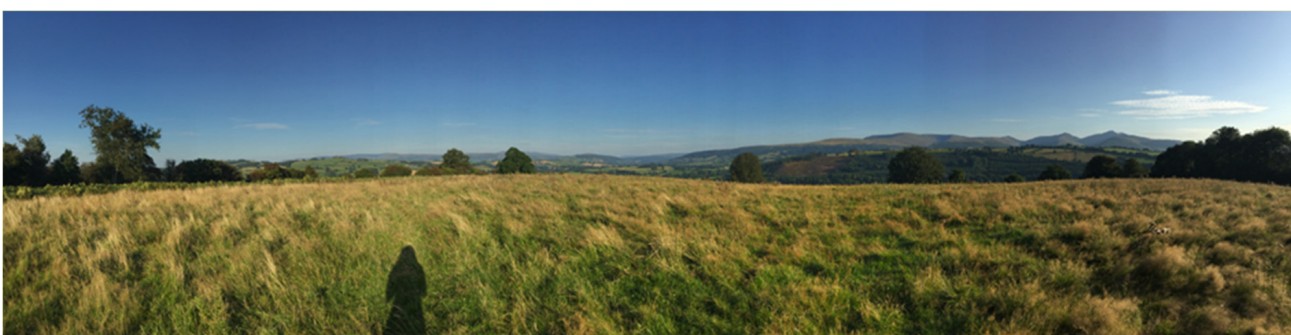

**Figure 1.** A picture of the space outside of Brecon in Slwch Tump, which is described as the setting of St. Eluned's shrine.

The shrine of St. Eluned was destroyed in 1698, and almost no traces of worship can be found on the spot today.[58] A pilgrimage map is provided, but I have found no traces of revival of these practices in the region.[59] Thus, to learn more about this tradition, we refer to text by Gerald of Wales:

> A powerful and noble personage, by name Brachanus, was in ancient times the ruler of the province of Brecheinoc, and from him it derived this name. The British histories testify that he had four- and -twenty daughters, all of whom, dedicated from their youth to religious observances, happily ended their lives in sanctity. There are many churches in Wales distinguished by their names, one of which, situated on the summit of a hill, near Brecheinoc, and not far from the castle of Aberhodni, is called the church of St. Almedda, after the name of the Holy virgin, who, refusing there the hand of an earthly spouse, married the Eternal King, and triumphed in a happy martyrdom.[60]

At first, we get to know in which part of the country the account is situated. Gerald of Wales also tells us that this particular region has a long history of Christian rule. The province of Brecheinoc is a sanctified area. This can be understood in two ways: firstly, that the king living and reigning there can produce holy women as his offspring, and secondly, that the grounds of this region had been transformed through the presence of many churches.[61] The particular church Gerald of Wales introduces to his listeners is guarded by a Holy virgin: St. Almedda. Also known as St. Eluned or Aluned, she was not only a virgin but, as the story goes, was martyred due to her quest of wanting to give her life to Christ.[62] Peter Brown has written extensively on the topic of how females in the early church gained both authority and freedom to break with patriarchal gender norms by rejecting the ordinary life of procreation for a life with Christ.[63] Caroline Walker Bynum further explains that, similar to chastity for men, virginity for women was almost a precondition for sanctity in the medieval period.[64] The text further emphasizes that Eluned was murdered in the act of protecting her chosen life of holiness.[65] This emphasis on martyrdom brings in an even stronger focus on the fact that hers was a life of sanctity and, thus, she had the ability to bring healing to others.[66] Similar stories identified in *Butler's Lives of The Saints*, further show that it was the task of the whole community to protect this kind of female vulnerability. If the community did not stand up to protect the holy woman who fled from matrimony into the arms of Christ, the space itself was cursed.[67] Clearly, the scenery is set to show the Glory of God in the region Gerald of Wales had entered.

The account of Gerald of Wales continues to explain how the feast he encountered was celebrated:

> [St. Almedda] to whose honour a solemn feast is annually held on the first day of August. On that day, a large concourse of people from a considerable distance make their attendance, and those persons who labour under various diseases, through the merits of the blessed virgin, receive their wished-for health. What, for me appears remarkable, is that at almost every anniversary of this virgin, similar events as these occur. You may see men or girls, now in the church, now in the cemetery, now in the dance, which is led round the churchyard with songs, on a sudden falling on the ground in ecstasy and silence, then jumping up as in a frenzy, and representing with their hands and feet, before the people, whatever work they have unlawfully done on feast days.[68]

What Gerald of Wales sees, presumably not just once as a curious exotic trait of his journey, but every year at the anniversary of the saint, he finds remarkable. People attend this feast from afar and are given healing for various diseases. One tenet of the *hermeneutics of suspicion* and *charity* is not to disregard aspects of lived religion and religious experiences of people living in another time and age than our own. This means that hagiographic materials and the depictions of the laity's worship practices are considered theologically relevant materials for discussion.[69] As Peter Brown has argued, particularly the cult of the saints incorporated many rowdy forms of practice that had become popular with the populous at the feasts of the martyrs, even when they were frowned upon by certain Church fathers. The experiences labelled as rustic or "vulgar" (both in ancient times and later overlooked by scholars) were even incorporated into the official liturgies of the Church.[70] As a continuation of this idea, I argue that dance historians need to become more nuanced in how they read and use texts by Church authorities. Too often, it is taken for granted that the Church condemned all forms of dancing, and then the depictions that are found of dancing are portrayed from the point of view of resistance.[71] Instead of looking into particular circumstances and nuances in the details of a story—as I attempt to do here—the diversity and multiplicity of understandings are lost.[72]

Contrary to the arguments that take for granted that dancing was forbidden in Church contexts, Gerald of Wales does not describe the dancing in *Itinerarium Cambriae* as problematic. Even when this kind of dancing leads to ecstasy and frenzy—potentially moving away from the pattern of sacred and highly organized dance movements—he does not judge the behavior of the dancers.[73] Even though elements of this specific event may fall outside of the "norm" of an anniversary of a saint, he finds them remarkable, rather than condemnable.

I find it particularly important to note that Gerald of Wales does not portray the singing and dancing described here as possession of an unclean *spiritus*.[74] In recent scholarship, when depictions of dancing are brought forth as part of the celebrations of saints, scholars tend to immediately assume that they are situations of possession.[75] Contrary to such understandings, Gerald of Wales's attitude here seems to instead depict Church leadership that understands that the *potential* and *praesentia* of a saint—when following the strict setup of ritual practices during a festivity—may have remarkable consequences, and do not necessarily indicate demonic activity.[76] The description of healings and dedication to the Blessed Virgin show another way of understanding the dancing, which I return to shortly. At this point, I mainly want to highlight the need for reading accounts like that of Gerald of Wales with a *hermeneutics of suspicion* towards dominant interpretations in secondary scholarship around dancing in and around churches in medieval Europe.

Kinesically, the description opens with a long row of people mixed across gender and age lines. This depiction breaks with the "normal" patterns of how church processions are to be conducted and what the rules of behavior between the sexes are expected to look like.[77] Such breaking of traditional social roles is precisely what Brown speaks of when entering the liminal space, not only of a cemetery or through a pilgrimage to a shrine (both

seem to be the case in this event). Particularly, Nancy Caciola's reading of this account emphasizes the importance of the cemetery as a liminal space, where encounters with the dead could be mediated through dancing.[78] In addition to such a view, I would argue that the possible liminality of this event lies in the fact that this is the feast day and liturgical period, when this particular saint would bring forth her *praesentia*.[79] Brown's account of how the festivities of the saints opened up for a "time out of time" could be one way of understanding why Gerald of Wales did not choose to condemn this particular practice of dancing.

Secondly, the long row of people winding their way in and out of the church and churchyard, and finally creating a round dance encircling (*circumfertur*) the cemetery, kinesically opens the possibility that, at the beginning of this celebration, the dancing and singing might have been more organized. In Gerald of Wales's condemnation of dancing in *Gemma Ecclesiastica*, he adds an *exempla*, where he explains that the problem with dancing on feast days was that the singing and the movements could distract the priest.[80] This is tied to the idea that if the priest made an error in the rite, it jeopardized the well-being of the whole community.[81] If we instead entertain the idea that this particular feast was one where the dancing was part of the ritual, such challenges for the priests would disappear. A small indication of the possibility of interpreting Gerald of Wales's writing in this way is the nuanced differences he uses in describing the singing. The word used for songs in *Itinerarium Cambriae* is *cantilena*. Watkins translates it as "traditional songs".[82] However, the term *cantilena* indicates a vocal song and has many usages, among which dance-songs, sacred songs and love songs can be found.[83] More importantly though, when Gerald of Wales condemns the singing on feast days in *Gemma Ecclesiastica*, he describes them as bad singing (*mala canentes*), referring to particular kinds of songs that were not appropriate. In contrast to this, in *Itinerarium Cambriae* he writes about dancing and singing without any indication of which kind, leaving the option that this is a completely approved form of singing for such an occasion.[84]

Furthermore, kinesically, the movements in the first part of the dance may suggest a more systematic form. There is no description indicating if the people are holding hands or not. Still, there may have been some synchronicity to these movements. Dancing and singing in a line and creating different formations are known from many folk-dance celebrations and depictions in Church art. The most famous of these are the two paintings by Fra Angelico from a completely different time and place than the feast of St. Eluned.[85] The visual of such a painting may assist us in seeing and sensing how more organized and worshipful movements would look like in the imagination of a medieval person (see Figure 2).

Part of Fra Angelico's *The Last Judgement*, commissioned for the Camaldolese Order (1425–1430), situated in Florence, Italy, depicts dancing in a long line through a garden. The movements start from the crowd of people being saved at the Last Judgement and move forward towards the heavenly gates of New Jerusalem by holding hands and dancing.

Another point of kinesic depiction in this passage is that the movements flow between a long line dance into the circle format. If there were enough people participating in the event, such a circle dance could have encompassed the whole shrine. Also, this type of dancing is found in artistic depictions. For a comparison between the two different forms, see the following two images (see Figure 3).

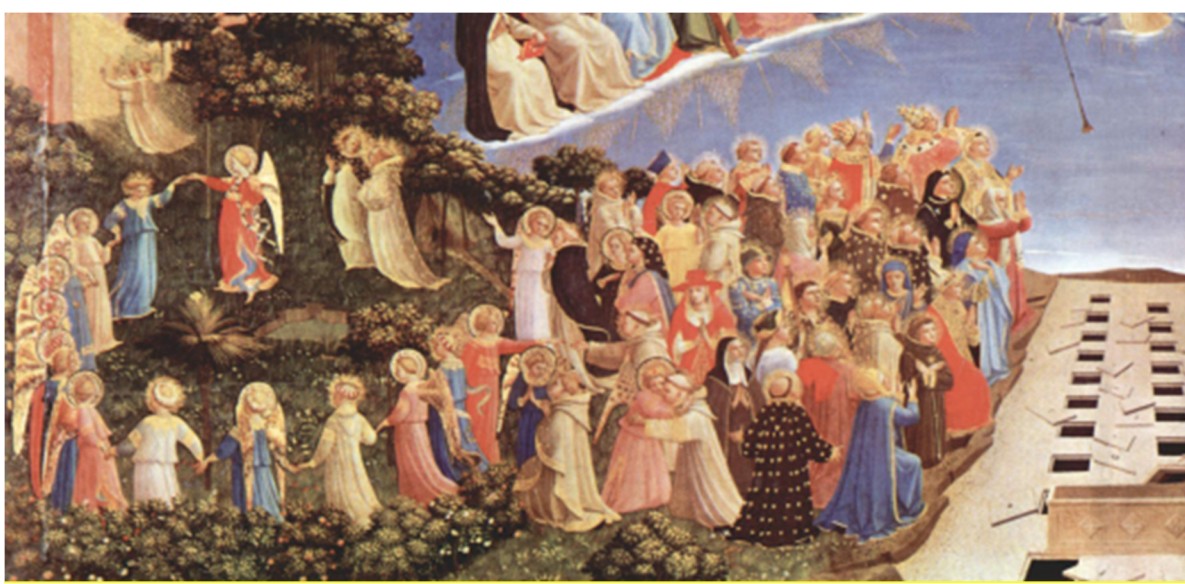

**Figure 2.** Fra Angelico was commissioned to do two different *The Last Judgement* altarpieces. This image shows parts of the piece which was commissioned for the Camaldolese Order (1425–1430) and is situated in Florence, Italy.

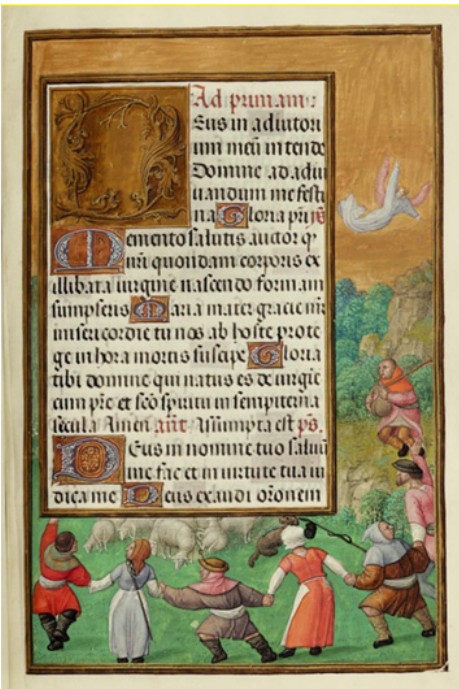

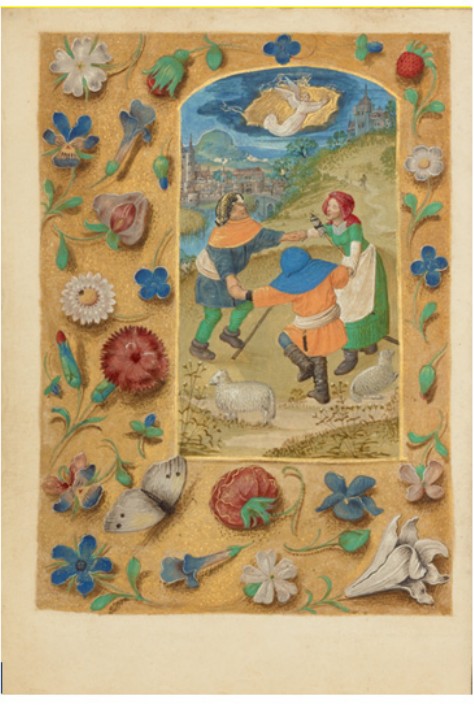

**Figure 3.** The image to the right is The Annunciation to the Shepherds by the Master of the Dresden Prayerbook or workshop, from about 1480–1515. The image to the left is The Annunciation to the Shepherds in the Rothschild Prayerbook 20, compiled between 1505–1510.

The image to the right is *The Annunciation to the Shepherds* by the Master of the Dresden Prayerbook or workshop, from about 1480–1515.[86] In this image, the male and female shepherds hold hands and dance in a circle. The image to the left is *The Annunciation to the Shepherds* in the Rothschild Prayerbook 20, compiled between 1505–1510.[87] In this image, the men and women form a long line of dancers similar to the saints and angels in the Fra Angelico painting.

As these images show, the concept of mixed-gender dancing, particularly in rural settings, is not foreign to religious art from the medieval period. Even when the recom-

mendations by church leadership would have opted for forbidding mixed-gender dancing, these kinds of images found their way into prayer books depicting worship during a liturgically important event of the year.[88] Thus, I suggest that imagining a more worshipful form of communal dancing during the particular feast we examine in this article should not be ruled out.

Comparing the kinesic descriptions in Gerald of Wales's story with the images found in the *Book of Hours of the Altarpieces*, from 400 years later and in a different region of Europe, there is reason to bring in some *hermeneutics of suspicion* towards the kinds of conclusions that can be made. There are, for example, aesthetic and practical reasons for why a depiction of dancing would be created in a line in the margins of the *Book of Hours*, while full-page pictures would more likely be depicted as a circle formation. So, while images in artwork cannot prove practices in the field, A. William Smith in *Picturing Performance* (Smith 1999) argues that it is unreasonable to suggest that depictions of dancing in paintings carry no bearing to something the artist has experienced and used as a model for their work.[89] Thus, I find it relevant to bring in a *hermeneutics of charity* and ask: could the first part of the depiction of the feast day of St. Eluned have been a worshipful initiation, or even a prayerful opening of the festivity? When we see that peasant and mixed-gender dancing are options for depicting prayerful worship in the *Books of Hours*, maybe it could have also been a practice that people embraced during a saint's feast? In the continuation of the story, we find further reasons for such a reading.

After the winding dance, Gerald of Wales describes that something abruptly happens (*subito in terram corruere*).[90] It is, of course, possible to understand the sentence to mean that the falling down and jumping up in a frenzy occur the whole time, while people are moving around on the cemetery grounds. Some earlier interpretations and translations that opt for the word trance in this section seem to lean towards this understanding.[91] There is much to say about such an interpretation.[92] However, this article format does not give room for all of those discussions. Rather, I want to focus on a reading based on the kinesic understanding described here, which has not been explored in previous scholarship.

Instead of perceiving the movements as all packed together in an ongoing continuum, we may pick out the details described by Gerald of Wales, and another reading opens up. He states that people suddenly fall to the ground. Reading this kinesically suggests a shift in the bodies and their movements. It is from a previous and different kind of motion that the bodies suddenly fall to the ground. What I thus perceive is a sequence of very different types of movements following each other. First, there is one kind of dancing, and then there is the falling. Furthermore, the exact Latin words used for what follows are: *extasim ductos et quietos*.[93] Therefore, I have not opted for the usage of language that refers to a trance, but instead, I have translated the passage in terms of ecstasy and silence. Shifting from singing and dancing in a pattern that encircled the churchyard to first falling, and then movements of ecstasy and silence, signifies several shifts in bodily tension and movement patterns. Yet, nothing indicates that this needs to be a trance.

Theologically speaking, ecstasy also signifies something different then trance. Ecstasy may occur when the Holy Spirit falls upon people.[94] Interestingly, it is this exact pattern that is described in Gerald of Wales's account. The dancers fall to the ground and then rise up in ecstasy and silence. Ecstatic forms of prayer are not foreign to certain schools of Christian contemplative prayer.[95] In his *De Poenitentia*, Ambrosius of Milan references events in St. Paul's life as dancing in the Spirit.[96] Also, medieval teachers of the Church have continued to speak about forms of prayer where the soul and body co-operate in a way that may lead to *extasis* or *excessus mentis*, which is an event where one moves out of the bounds of normal bodily being.[97] There are, thus, significant cues in Christian traditions to interpret the dance of the feast of St. Eluned as a situation of worship where people are affected by the Holy Spirit. The bodily gestures described by Gerald of Wales of what might be happening do not fall out of context, and they are not automatically condemnable. His descriptions further indicate why this is not a situation of possession by an evil spirit

that is then cleansed by the saintly *praesentia*. Instead, the movement vocabulary shows prayerful invocations that lead to the arrival of the Holy Spirit.[98]

It is only after this passage of ecstasy and silence that Gerald of Wales describes a sense of frenzy (*phreneaim raptos*). Kinesically, the progression goes from dancing and singing to falling and reaching an ecstatic silence. Immediately after this, people move into rapture and leaping up to represent different movements.[99] Instead of being in a continued state of trance, there is a clear progression with various movement patterns ending in rapture. Kathryn Dickason points out that the terminology of rapture had a very broad set of meanings in the medieval period. It could range from abduction and seizure to ravishment and rape.[100] However, both Dickason and those she builds her argument upon seem to have overlooked that rapture of a spiritual sort, just like prophecy, induced by singing and playing music, are also part of the stories of the Bible.[101] Not only did David leap up and dance in front of the Ark, but King Saul fell into a frenzy when the Spirit came upon him in his anointment to become the new king.[102] Furthermore, as already indicated in Ambrosius of Milan's writing, the Apostle Paul speaks about being seized by the Spirit and transported to an altered state of consciousness in his letter to the Corinthians.[103] That passage is sometimes referred to as a rapture, not only ecstasy.[104] These examples show that the idea of worshipfully engaging in music and dancing, which leads to the arrival of the Holy Spirit and movement patterns of people taking an extraordinary form, could very well be part of something Gerald of Wales found remarkable, even a blessing! Once the Holy Spirit, also called a Spirit of Truth,[105] arrives, we see that Gerald of Wales interprets this as a moment where earlier hidden aspects of people's lives are revealed in the movements they leap up to represent.[106]

Let us now move to the next section of the story, which depicts what happens in more detail, once what may have been the Holy Spirit appears on the scene:

> ... you may see one man put his hand to the plough, and another, as it were, goad on the oxen, mitigating their sense of labour, by the usual rude song: one man imitating the profession of a shoemaker; another, that of a tanner. Now you may see a girl with a distaff, drawing out the thread, and winding it again on the spindle; another walking, and arranging the threads for the web; another, as it were, throwing the shuttle, and seeming to weave. On being brought into the church, and led up to the altar with their oblations, you will be astonished to see them suddenly awakened, and coming to themselves. Thus, by the divine mercy, which rejoices in the conversion, not in the death, of sinners, many persons from the conviction of their senses, are on these feast days corrected and mended.[107]

Kinesically analyzing this passage, one can sense a whole range of gestures pertaining to the agricultural tasks and labor of the populous portrayed. The tasks described are gendered, with work related to men first and those of women or girls coming later. In a note, we can also read that the rude song referred to may have been a Welsh ploughing song, that made the work of both oxen and men more rhythmic.[108] Gerald of Wales further explains that these gestures are brought to the altar as an oblation or offering (*oblationibus*). Once the people arrive inside the church, they suddenly awaken and come to themselves.

Finally, he also writes that those who took part in this gesturing, and those who have seen and heard (*vivendo quam sentiendo*) what occurred, correct their way and are mended. This last part is quite tricky to translate as the word *ultionem*, deriving from *ultio*, may indicate everything from avenge, revenge and punishment to indulgence.[109] It is unclear if the text refers to setting free those who had indulged in their sensory experiences, meaning, "whatever work they have unlawfully done on feast days", or that they were set free by taking part in the gesturing. Another plausible explanation can be derived by considering those watching the gesturing. Was it their indulgence in seeing and hearing the gestures that freed them from the punishment that otherwise would have fallen upon them?

Carl S. Watkins is one of few authors who goes into a lengthier discussion on how to understand this passage based on the whole account. He states that what is acted out

in this situation is a local response to the Church's teaching about sacred time and sin. He suggests that the members of this community had absorbed the idea that working on feast days was sinful. Through their movements, they signaled that they were nevertheless averse to this teaching, and thus perhaps pressured to seek absolution. As there were no readily fulfilling penances, local rituals had emerged to lighten the lot of post-mortem punishment. Watkins concludes this is a ritualized re-enactment of the sin and seeking intercession with the saint.[110]

Watkins seems to come to this conclusion based on three different clues. First, he sees that Gerald of Wales approved the ritual as "appropriate". The approval is shown through the priest being placed at the heart of the ritual, dispensing absolution at the altar rail to all those who had participated in the rite.[111] This is an interesting understanding, as the text does not speak about priests meeting the afflicted people at the altar. It only states that people were "brought into the church and led up to the altar with their oblations or offerings".[112] Kinesically, the text offers another possibility: the worshipers present at the shrine assist the afflicted people in entering into the space where St. Eluned showed her *praesentia* and *potentia*. Invoking the *praesentia* and *potential* of the saint alone is what brings people back to their senses.

In some biblical passages, such as 2 Timothy 2:15, Colossians 1:21–23 and Romans 12:1–2, it is claimed that a sinner is only required to present themselves in front of God to receive forgiveness and new life.[113] There is a link between sins leading to death in these biblical examples, but there is no mention of post-mortem punishment. Instead, worshipfully presenting one's body in front of God is a practice that needs to be renewed repeatedly.[114] Furthermore, such an offering for the absolution of a particular sin does not require a priestly presence. At the same time, Brown tells us that accounts of healing at the shrines of the saints are always a communal act.[115] From this point of view, presuming priestly guidance on this occasion may be apropos. Nonetheless, it is crucial to notice that nothing is said about exactly *how* the offerings of the dancers were met, or by whom.

Watkins's second point emphasizes Gerald's explanation of sin being at the core of this ritual.[116] At the end of the account, Watkins takes the statement, "God in mercy does not delight in the death of a sinner, but in his repentance: and so, by taking part in these festivities, many at once see and feel in their hearts the remission of their sins, and are absolved and pardoned",[117] as a key moment for understanding the ritual. He reads this as pinpointing that not God alone, but the priest too, should have an attitude of forgiveness towards sinners. Furthermore, it seems to be the statement about death which leads Watkins to speak about a ritual for absolution from post-mortem punishment. Even though asserting a link between sin and death is understandable, I do not entirely agree with the amount of emphasis Watkins places on post-mortem fear. He writes that there was much preaching concerning the need to make "full satisfaction" for one's sins in this specific period.[118] Yet, as preachings on purgatory and an equal emphasis on the punishments of hell were only emerging at this time, I find this statement somewhat arbitrary.[119]

Instead, one could read the account as a more general question about healing and re-entry into the community. Those who were healed, when participating in the ritual of the feast day of St. Eluned, were not only those who had sinned against the Sabbath rest by working. In the beginning, it is stated that "persons who labour under various diseases, through the merits of the Blessed Virgin, received their wished-for health".[120] This opens up the possibility that the healings were from many other afflictions, not only those related to the suggested actions breaking God's commandment of rest. Brown has argued that the festivities of a saint day celebration were periods when God's acceptance of the whole community could be sensed and seen by bringing disparate members together.[121] Speaking about the miracles that happened during the festival of Saint Martin, he states:

The barriers that had held the individual back from the *consensus omnium* were removed. "With all the people looking on," the crippled walk up to receive the Eucharist. The prisoners in the lockhouse roar in chorus to be allowed to take part in the procession, and the sudden breaking of their chains makes plain the amnesty of the saint.[122]

So, even when ecstatic dancing, frenzied movements and gesturing may have been expressions shown only in the bodies of a few members of the congregation, the whole community was affected by what they saw and experienced. One could also take part in the miracle and in the celebration of the mercy of God as an onlooker. This renewal of communal membership through the yearly festivals were a praxis that created solidarity and communion between people from different social classes and positions.[123]

I find the final point that Watkins makes to be very relevant, though for a slightly different reason than he does. He makes a more general note about the 11th and 12th centuries being a period during which local customs could be particular and specific. On one hand, Watkins writes that "universal" teachings of the Church on questions of penance, the eucharist, good works and last things were still being formalized by both monks and scholars. However, there was no existing structure explaining exactly what the unified body of the official teachings of the Church would be. At local parish levels, the community identified themselves with their local priest and the community gathering for worship. This meant that there could have been specific rituals that were developed in one particular community and practiced only there.[124]

Vincent Lloyd explains that rituals are actions that reinforce social norms.[125] He writes: "Ritual is understood as a community practice, reflecting or growing out of social norms".[126] This local practice could thus have been something that Gerald of Wales found worthy of promoting. It bound the community together and taught the people who arrived at the feast of St. Eluned important lessons about Church teaching. The way that Gerald of Wales describes this event indicates a ritual that showed God's mighty *praesentia* and the merciful acceptance of those who chose to present themselves under the Church authorities. In such a view, the ritual itself created what Rohmann calls the liminal space of being in between perdition and salvation.[127] Dancing—with its ambivalent status—appears in such a story as an indicator of liminality, or the consequence of liminality. When emphasizing the idea of specific and local rituals as an expression of religious life in the early medieval period, dancing and frenzied gesturing—on their account of being indicators of liminality— are not threats to the Church authorities. Instead, they become tools that can be used to bring people together. Thus, reading the account of Gerald of Wales with liminality as a theoretical starting point is one way to understand why he did not condemn the practices, and why the community was allowed to celebrate their saint in a manner where singing and dancing were part of the yearly festivities.

Choosing this route limits one's understanding of the Church and Christianity to a male and patriarchal structure. It also strengthens a tradition that is built on reinforcing norms and upholding hegemony.[128] Such an option is not the only path forward. Thus, the second part of this article turns to the vital critique of liminality presented by Caroline Walker Bynum.

## 5. Whose Story? The Critique of Liminality

Watkins introduced the idea of reading the account of Gerald of Wales as a local and "place-bound" practice. This raises the question of what stories we accept as the most important narratives for understanding this dancing event? Are we looking towards theological or philosophical expertise at the educational centers of Europe to understand the praxis, or do we turn to the less known manuscripts of local writers? As scholars and interpreters, do we center the more subtle voices of practitioners and laity found in artwork, praxis, legends and unofficial records or certain textual statements by Church authorities in the region? Watkins presents these challenges when he speaks about the

differences in understanding that arise if one listens to cloistered teachers, such as Abbot Ailred, and compares his statements to those of a leader of the local priests, like Gerald of Wales.[129] Simultaneously, these approaches center on male voices and more educated people. More importantly, they often take the elite male experiences as an implicit norm for all expressions of living in a Christian tradition. What would happen if we instead looked at the dancing from the point of view of the laity, and the women who took part in the dancing? These questions, and the critique of liminality that Caroline Walker Bynum has expressed, go hand in hand. In this second part of the article, I place the description of Gerald of Wales, as well as the readings of it by Carl S. Watkins, in the foreground and adopt a less text-bound approach towards the feast of St. Eluned and her followers.

In her *Fragmentation and Redemption* (Bynum 1991), Walker Bynum explains that a challenge with Turner's theory on liminality is that its emphasis on reversal and elevation only makes sense in a world of hierarchies, and for those individuals (male, aristocratic and educated elites) who have the real possibility of climbing a social ladder. In short, Walker Bynum explains that Turner's ideas about social dramas could be read—in the episodes of a single saint's life—as a succession story from "ordinary" life to a "conversion" event where the (male) person passes, either through reversal (turning into woman, bestial or fool) or inversion (claiming poverty or female subjugation), into a climactic episode at which reintegration happens, and then, the individual lands in a new state of being. In the example of a dominant symbol, the idea of reversal or inversion happens when a person takes part in a ritual action that questions the dominant structure. The dominant symbol supports a person in their transformation and elevates them from one status to another.[130] However, the prerequisite for this is that the person has the freedom to move between different forms of status, or that there actually is a kind of divine intervention which changes reality.

In contrast, Walker Bynum's research on females in the medieval period shows that these social dramas and conversion patterns, or dramatic changes, cannot be found. Instead, her investigations of stories about women (especially when told by women) describe patterns of continuity and the deepening or enhancing of an experience. She further argues that when stories, where the experiences of laity or women are not immediately perceived as liminal, are examined *with* the experiences of these people in mind, something else than liminality comes to view. What may have been a release or escape from ordinary life for those in a status position, is not so with those who are powerless. Instead of voluntary poverty, weakness, or nudity in the process of *imitatio Christi* for a privileged person, the unprivileged individual does not turn to wealth, strength, or splendor in describing their process (as a reversal), but to a path of continuous struggle.[131]

From the male-dominant point of view, following Walker Bynum's analysis, the pilgrimage to St. Eluned's shrine provided ample opportunity to release societal structures and live in a "time out of time". In the eyes of a male clerk like Gerald of Wales, the partaking in frenzied dancing may very well have shown the power of a female saint and intimate interaction with a(n idealized) Virgin, which Walker Bynum explains to be characteristic of many medieval males seeking a mystical path. Finally, the story of St. Eluned's life, framed in the telling of Gerald of Wales, is one of "high" romance: she has to die violently to become united with her beloved Christ.[132] What appears if we turn to the daughter of Brachanus herself, or the community of worshippers around her? How would they have expressed their understanding of what happened in the dancing? Does the dancing create a different kind of theology?

One of the ways to go about this is to turn away from Gerald of Wales's focus on sin and redemption and read with *suspicion* what can be found "between the lines" about the female characters of the story.[133] The first relevant passage for such a reading is: "The British histories testify that he had four and twenty daughters, all of whom, dedicated from their youth to religious observances, happily ended their lives in sanctity".[134] Even though this statement was not penned by the hand of a woman, it does testify to Eluned not being a lone woman. She had a community of sisters around her. Using the *hermeneutics of suspicion*

towards dominant male descriptions, and looking with a *hermeneutics of charity* for the lived religion of the women in the story, another possible perspective of this feast day is visible.

The text continues by explaining that Eluned refused the hand of an earthly spouse. From this small detail it is plausible to imagine that Eluned, like so many other medieval women, experienced her relationship to Christ from the point of view of being his bride. Walker Bynum explains:

> Medieval women, like men, chose to speak of themselves as brides, mothers and sisters of Christ. But to women this was an accepting and continuing of what they were; to men, it was reversal. Indeed, all women's central images turn out to be continuities.[135]

Being the bride of Christ displayed the sense of continuity already mentioned where, for females, the active engagement with God did not turn their world upside down. Instead, they continued as women in undertaking very female tasks (being sisters, mothers and spouses). Further, Walker Bynum affirms that the image of a bride or lover of Christ was not one of passivity, but rather an idea of an active involvement and a very sensual engagement with God.[136] We do not know how Eluned experienced her sense of belonging to Christ, but the account does tell us that she was ready to break the worldly marriage engagement her father had made on her behalf and flee from the safety of her home.

Furthermore, Eluned was not alone in this experience of leaving the "worldly" life of marriage and caring for children in order to hide out and "elope" with Christ as her bridegroom. Both Jane Cartwright and Liz Herbert McAvoy argue that some of Eluned's sisters lived anchorite lives.[137] In *Anchorites, Wombs and Tombs—Intersections of Gender and Enclosure in the Middle Ages* by McAvoy and Hughes-Edwards (2005), it is described that choosing the anchorite life of seclusion and solitary prayer seems to have been a very popular option for women, especially in the context of the British Isles.[138]

The work of McAvoy and Hughes-Edwards traces the development of a solitary life from the Desert Mothers and Fathers of the early Church into the Middle Ages. It particularly looks into the phenomena that developed and guidebooks written for men and women in a Northern European medieval context.[139] The study shows that the first written document of a guide for those who wanted to choose the solitary life in an English context already existed in 1080.[140] This means that during the time of the written description of Gerald of Wales, the tradition of female recluses was already established in the proximity of this region. It further means that even though there is no written record of the *Achrene Wisse* or a similar text found in Welsh, the path of an anchorite life might have still been a possible strategy for Eluned. More specifically though, reading about Eluned's life would have been a story that created associations to an anchorite life in others.[141]

McAvoy and Hughes-Edwards describe that, even though the idea of an anchorite developed out of the mix of the image of a monk withdrawing into the desert in search of a solitary life, and the suffering of the more urban martyrs, the anchorites and hermits of Northern Europe were an entity in their own right. The hermit, according to Herbert McAvoy and Hughes-Edwards, developed from the idea of withdrawal into a solitary life of prayer and seclusion. In the consequent Northern European tradition, however, this came to indicate a person who was ideologically solitary and, nonetheless, free to move about physically. The anchorite, in contrast, was a person who chose to withdraw from the world, but this withdrawal could be achieved in a solitary form or in a community with others.[142] In the Northern European tradition this came to indicate a person who found their place in the "bleak and isolated islands, wild, impenetrable forests, perilous, boggy marshlands".[143] Thus, the main difference here was that these individuals were confined to a stationary, fixed position.[144] At the same time, such a life did not always mean seclusion from other people and communities. The female anchorites, in particular, often tended to have a rich life of social connections, with people visiting them, engaging with their teaching and spiritual guidance. Sometimes they even rose to prominent political positions in their society.[145] Thus, when we read that St. Eluned chose a life of hiding, as the bride of

Christ, this should not be understood as an insignificant step into isolation. She just as well may have been aspiring for a space of prayer and seclusion—a deepening of her life with Christ—while simultaneously enhancing her status as the daughter of a prominent leader. This kind of contextualization of St. Eluned's life, and her community of females in the region, is more important than what might be clear at first glance. It brings the depiction of the feast of Eluned away from the focus on a ritual, where Church authorities have the last say in how the celebrations evolve. Instead, we may be looking at a female role model who spoke to women and laity in the local community, and strengthened the personal connections between individuals and the experiences of the divine.[146]

When I visited Brecon, I learnt that the Cathedral Church of St. John has also been in connection with a Benedictine priory.[147] In their story of themselves, they state that Eluned visited the monks in their community before heading up to the hill, where she was murdered. It is stated that these brothers offered her shelter. She received their kind offering. Yet, after some time, she continued her journey and was subsequently found by her suitor. We will probably never know why she did not stay with the brothers in the priory, or why she continued up into the countryside. Could she have been looking for this more solitary form of life? Her story allows for the possibility of reading her life as a devotion, and that she did not want to give in to the patriarchal structures of the normal society. This is not a climbing of a social ladder, nor is it a transformation of one's position, but a constant struggle with the dominant powers of oppression.

Vincent Lloyd tells us that the life of sanctity is an ambivalent path. On one hand, the political potential of sanctity lies in acting as if there are no norms.[148] The choice of breaking with the norms of what was expected of a woman and fleeing into the arms of Christ may have been a strong statement of walking on an unthreaded path. Sanctity as a strategy may act to break the hegemony of the visible.[149]

On the other hand, when one makes sanctity into a lifestyle, it inevitably gives rise to norms. It tethers the plane of practices to the norms.[150] This is why it becomes problematic when we only read the stories about female saints penned by male authors in the Church. What once may have been a strategic path of breaking norms is quickly turned into a new hegemony of expected behavior. Still, there are ways to unearth the female voices and find how their stories differ from that of the male practices.

Jane Cartwright, in *Feminine Sanctity and Spirituality in Medieval Wales* (Cartwright 2008), explains that the statements about Eluned's sisters is not merely a remark that was put into the story to accentuate her father's goodness as a king, or to emphasize the Christian character of the region. Each of the sisters were venerated in the region, and one of them is even described as creating a bloodline directly to the male patron saint of Wales, St. David.[151] The most common trait of these sisters in the stories is that they, just like Eluned, wanted to become brides of Christ. Along the lines of Walker Bynum's observations of female saints finding their calling either early in life or rather late, all of these women are told to have started their walk with God at an early age.[152] None of them are described as having had dramatic conversion stories. Still, when they take a stance for their virginity in puberty, they are either murdered by pagan Saxon kings wanting to marry them or, like St. Melangell and St. Gwenfrewy, they are miraculously saved from the men pursuing them so that they can start a community for female followers.[153]

Cartwright further describes a clear difference between the stories told about the male and female saints in Wales. In the Welsh context, the male stories often follow the narrative pattern of a secular hero's journey.[154] This is a different pattern than the liminality described by Walker Bynum. Nevertheless, the male stories follow a clear social drama. Their childhood is great. They receive a good education, and then they "come out" by performing a miracle at some point. After this, they live a long and peaceful life, healing many and punishing unbelievers.[155]

In contrast, the female stories all involve scenes of abduction, torture, murder and even rape (to explain the birth of a male saint). In conclusion, virginity and martyrdom or life-long dedication to asceticism became the pattern for females in Wales to reach

eternal salvation.[156] Importantly, when looking at accounts like these, is that the male gaze may subscribe a certain amount of high drama and violence to a story where the female "protagonist" sees herself as only following a continuous path with Christ, going from the role of sister to that of spouse and mother.[157] Thus, the theme of continuous struggle, portrayed by Walker Bynum, can still be a dominant trait in these stories. What is clear at least is that the women do not have a moment of "coming out", nor do they "live happily ever after", creating miracles and showing up as a male leader.

It is mainly after their death that the female saints of this region become sources of healing, and create a following through the celebration of their feast days with pilgrimages to their shrine, holy well and/or statue.[158] Many are also honored in poetry and songs, while some even get a play of some kind performed during their celebrations.[159] Also, in the stories told about St. Eluned, we may find that, not only did her site of martyrdom become a place of pilgrimage, she was decapitated, and at the site of where her head fell, a Holy spring burst forth.[160] Herbert McAvoy explains that the way that the feast of Eluned is described, as a celebration taking form, not primarily inside the church where her relics were contained (according to Cartwright), is suggestive of the space itself having been sacralised.[161] Could the dancing also have been experienced by those who took part in this celebration as a path for creating a theology of sanctification?

Cartwright tells us that the shrine actually contained Eluned's relics; this is a further clue to how we can understand the dancing. The kinesic depictions of a progression of different forms of dance that I suggest in the first part of this article, may now find a slightly altered nuance. The dancing in the beginning may have been an opening and prayerful greeting of the saint, and the more gesturing movements may have been a way to portray to the saint where various afflictions laid. In the end, it was the true *praesentia* and *poetentia* of the saint—evoked by the praise—that brought forth the healing. Her miracle-working power and the particular celebrations around her shrine may even have had such far-reaching importance for the region that Gerald of Wales framed it within the language of sin and repentance to bring it under Church authority. The people celebrating may have had a different story to tell.

Stated in another way, the importance of a sanctified life does not mainly lie in what happened in her actual life. Instead, it lies in what her life and death gave inspiration to. When the life of sanctity was defined by Lloyd as living as there are no norms, he describes that sanctity may inspire liturgy. While rituals are described as practices that uphold norms, liturgy is a practice that has the potential to alter norms.[162] "Liturgy involves specific, exceptional practices in which one acts as if there are no norms".[163] It aspires to create a gap towards the norms in order to alter them. This is done partly by offering a foretaste of a world to come.[164] So, when the life of St. Eluned speaks about possibilities that are unavailable within the norms, and when listening to stories about her life leads to people taking up new ways of being, altering the way they treat each other, themselves and the creation around them, one can say that Eluned's life has laid the foundations of a liturgy. When a community comes together to celebrate this life, and such celebrations contain elements of exceptional practices that lack ordinary norms, we may say that her life has created a liturgy. Her life has been able to loosen the forever-present pull that social norms have on us, and thereby has broadened people's political imaginations.[165] This broadening of imaginations may have also been something that the Church authorities found scary and, thus, they wanted to steer the reception of her story away from a living liturgy and into the form of a ritual, which can be contained.

Interestingly enough, not only did the yearly feast of St. Eluned gain a following, so did her life. Cartwright tells us that she was able to localize a community of nuns founded at St. Eluned's shrine in Usk, sometime before 1135. Cartwright bases this on the work of William Worcestre (1415–1485), that states that the convent was founded by Richard de Clare (c. 1153–1217).[166] When Gerald of Wales wrote his account, therefore, an established nunnery probably existed on the site. Communicating St. Eluned's story may have inspired

more women to follow in her footsteps of a sanctified life. Further, it may have been a story written at least partly with the intent to support these sisters' lives.

Cartwright further tells us that very few convents with women were able to establish a prosperous and strong following in Wales. This was partly due to the Cistercian order being favored by the Welsh nobility; they were not as willing to establish female houses as the Benedictine communities found in England.[167] It is thus curious that the nuns that Gerald of Wales may have been promoting with his telling of the story of St. Eluned's feast day were Benedictines.[168] He might have had a double interest in establishing a Benedictine house (tying the Welsh family to the English network) and strengthening a newly founded community with the revenues gained from pilgrims. Whichever way, the community of nuns would have played an integral role in the celebrations of the feasts, and by writing about it, he supported both the ritual and the liturgy established around her shrine.

Brown tells us that the shrines of the martyrs became places where those who had been touched by the saint gathered on a regular basis. He writes that the laity, especially women, were the ones who cleaned the shrines, cared for the distribution of food, and welcomed the pilgrims to the holy places.[169] At later periods of time, these tasks were often taken up by more established communities when such could be formed. It was not only the task of the community to care for the shrine and the pilgrims, it was often also a joint task of the laity and the religious "professionals" to build a liturgy around the festivity.[170]

## 6. The Liturgy of the Feast

Taking the stance that what we have read in the depictions of Gerald of Wales at least party describes the liturgy practiced on the feast day of the saint, new possibilities of interpretation open up. First of all, it seems more likely that the singing and dancing in the cemetery, inside and outside of the church, were truly signs that all of this area was sacred grounds.[171] The healing power of the well and the presence of St. Eluned's relic opened up for rejoicing as if there were no norms.

Brown tells us that an important part of the understanding of relics was that the healing potential existed in the tension between the saint now resting in the calm and delightful peace of afterlife, and the immense pain they had been able to endure in their bodily existence on earth. This *poetentia* and *praesentia* of the saint arrived when the community gathered together to "remember" and welcome her. This was often done by reading the *passio*, a story of the sufferings of the saint.[172]

Returning to the stories found about Eluned, Cartwright tells us that, unfortunately, none of the Welsh saints appear to have been canonized by the Roman Catholic Church. There are also very few official *vitae* or other written documents that give full accounts of the whole lives of these local saints and the cult of their veneration.[173] Therefore, we do not know what kind of a *passio* was being read at St. Eluned's feast day. It may even be the case that Gerald of Wales's account would later be used as a *passio*. On occasions like these, Felski's words about the limits of a critical stance in understanding our materials may give way to what Seeta Chaganti calls the invention of imagination; the need to let go of our claimable facts and play with the scant findings we do have.[174] Re-constructing history from the female and danced point of view requires more openness to what can be found between the lines of what is told in texts and archives.[175]

Instead of building the story of St. Eluned from her non-existing *vitae*, there are other ways to bring the celebration of her life into focus. Cartwright describes that, when it comes to other Welsh female saints, there are strands of poetry, pictorial narratives or historical records that can be used instead.[176] Bringing many different strands together may create a fuller picture. This raises the question: could the singing and dancing described at the beginning of St. Eluned's feast be seen in a similar light? Did the singing contain a local song or poem written in her honor? Was the gesturing a way to show affliction in one's body, that then activated the relic, just like reading of the suffering in the *passio* would?[177]

Cartwright tells us again that, in the case of Eluned, there is neither poetry nor records of her feast day found in the Welsh calendars. She turns instead to the account of a very similar story to that of Eluned, that of Gwenfrewy, probably the most renowned Welsh female saint. By examining their lives side by side, Cartwright attempts a reconstruction of Eluned's life that is denser than that in Gerald of Wales's version.[178]

The story of St. Gwenfrewy states that she refused sexual intercourse with a local prince and therefore was decapitated. One account tells that St. Beuno resurrected her, and when he placed St. Gwenfrewy's head back on her shoulders, a Holy well sprung forth from the pool of the virgin blood spilt on the ground. This well was said to have unusual stones stained with blood at its bottom, and to be blessed with healing powers.[179] The symbol of blood and a sanctified spring associated with miracles are, thus, joined together in this female *vitae*.[180] Furthermore, the miraculous healing powers of her well attracted both pilgrimage and festal commemorations from the laity, as well as from people in positions of power. These celebrations and popular hagiographical collections, including her story, circulated in the southern parts of the British Isles from 1138 to 1690.[181]

As we can see, many of the elements of these stories are the same. There is the beheading and the emergence of a holy spring. There is a life of suffering, and healing given to those that visit the space. In biblical records, we may also find a connection between important female characters and their ministry taking form as a practice in a city's outskirts and close to water.[182] Furthermore, raising the question of lay celebrations and healings happening at a Holy well also brings up the question of what, if any, role priests and clergy played in these celebrations. What if the liturgy itself—the specific, exceptional practices of St. Eluned's feast day—were a joint effort by the nuns at her shrine and the people who came for the celebration? What if they had—in their dancing—formulated a theology of the feast (as Ricoeur suggests art can do)?

Strengthening the possibility of these ideas is also the fact that particularly female lay activities seem to have been an important part of the way the Welsh community was structured. One of the reasons that Cartwright gives for there being very few nunneries in Wales, is that the dowry and laws around heritage made it quite impossible for women to own land and, thus, establish nunneries. It was also more complicated for them to choose to go into religious orders without strong financial losses to the community, or the family from which they originated. What happened instead was that women, from their positions as wives and widows, invested their movable goods into different forms of charity and worship services.[183] We already know, from Cartwright's research, that this meant that many of the local veneration practices driven by women centered around themes of motherhood and married life.[184] This raises the further question that maybe the liturgical practices established around the festivities of women like St. Eluned were a collaboration of the needs and wishes of the laity and women in the community. Perhaps there was not even a reading of a *passio* established in this liturgy. Instead, the festivity and the liturgy around it may have arisen more in continuity with the need of the people to gather in thanksgiving and commemoration. Maybe all of the gestures of different types of work were not done in a frenzy, as Gerald of Wales suggests. Maybe they were instead movements that created thanksgiving statements over the gifts that St. Eluned, sanctifying life and death, had poured over this specific space and landscape. If this feast day was a local custom, created by the female community gathering around the shrine of St. Eluned, it could very well be so that the laity and women of this space had their very own connection to the importance of St. Eluned.

This last suggestion was a completely new idea that came to me when I visited St. John's priory in Brecon. I was struck by the name of the chapel where St. Eluned is depicted (see Figure 4).

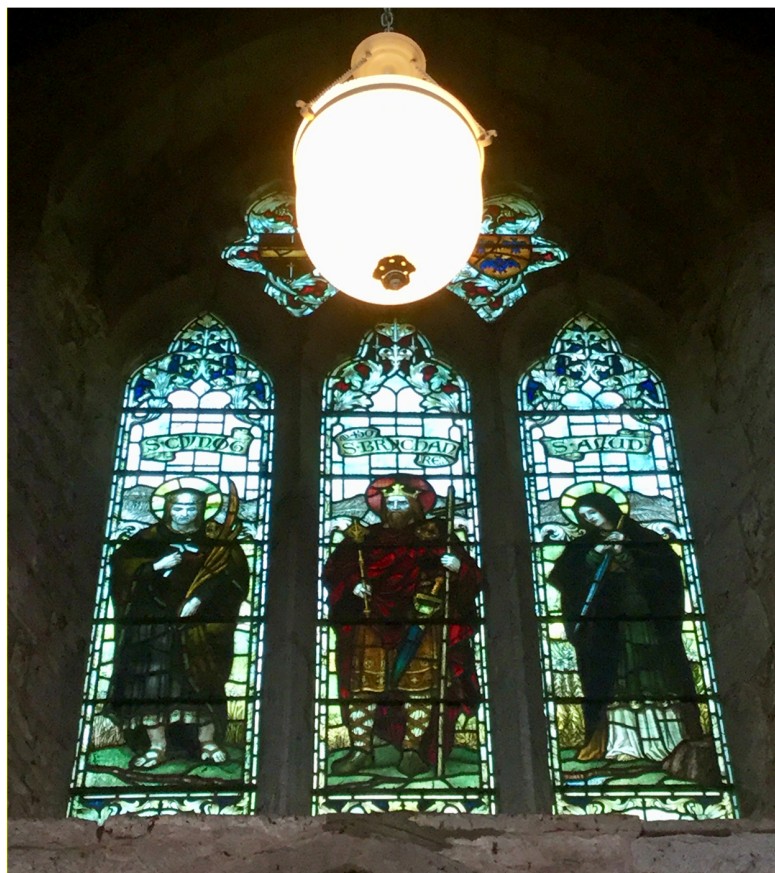

**Figure 4.** Stainglass window in the Cordwainers chapel, in Brecon Cathedral. Depicting to the right St. Eluned, in the middle, her father Brychan and to the left, her brother, Cynog. Image taken by author during a visit in 2019.

In the image, we can see Eluned on the right with a sword in her hand and a stone at her feet. In the middle stands her father, Brychan. He wears a cloak fastened by a penannular brooch, carries a sword, and is crowned. He also holds a sceptre and a cross. Finally, to the left, is her brother, Cynog, holding a martyr's palm, a cross and his famous torc. Further, her brother has a Celtic tonsure and wears a similar cloak to that of her father. The depictions were created and installed in 1911, so they do not have a direct relevance to the period in focus in this article.[185] However I learned a curious detail about the chapel where these images of St. Eluned and her relatives are located: it is called the Cordwainers chapel. When searching for more information about the meaning of that name, I found out that, in Brecon Cathedral, the Cordwainers (shoemakers), the Skinners (leather workers), the Tuckers (fullers) and the Weavers all had their guild chapels in the aisles of the Priory Church.[186] These are the very trades portrayed in the story of St. Eluned's feast day through the depictions of gestures. I have not been able to pursue this thread any further. However, it indicates that these guilds may have had a special relationship with the story of Eluned at some point. Such a connection could open up the possibility that the wild gesturing portrayed in the story of Gerald of Wales, either in the instigation of that liturgy or later on, indeed may have grown into carrying completely different meanings than movements depicting sinful affiliation. The naming of the chapel and placement of St. Eluned's image in its window is a mystery requiring much more investigation. Simultaneously, they stand there as a reminder that the lived religion of the laity and the religious experiences of the women may not leave as clear marks in history as the dominant narrative of male clergy do; still, they cannot be erased entirely. Pursuing the stories from the margins and from marginalized voices may need other tools and theories, yet it is an endeavor worth engaging in.

## 7. Conclusions

This article takes a look at the story of the feast day of St. Eluned depicted by Gerald of Wales in *Itinerarium Cambriae*. I first show that many theologians and historians who engage with stories from the medieval period that depict dance carry an assumption about the problematic nature of dancing in, or close to, churches. With such a pre-set lens on the dance practices and descriptions, particularly more rowdy forms of dancing, are seen as something that needs to be "explained away". With the tools of a *hermeneutics of suspicion* and *charity* and a kinesic approach towards historical materials and secondary sources, I turned towards the feast day of St. Eluned. My first aim was to show that this particular dance event was not considered by an authority of the Church, like Gerald of Wales, as something he needed to condemn. With the help of Peter Brown's use of Turner's theory of liminality, the dancing at the feast of St. Eluned can be understood as a remarkable, yet completely acceptable, course of events during the celebration of a saint. This is a different take from earlier accounts of this story, where the liminality of the cemetery or dancing as an ambiguous practice is in focus. Contrary to such readings, my first claim is that dancing may have been an acceptable form of worship in the medieval period. Furthermore, when the dancing turned into more ecstatic formats, and Gerald of Wales describes the gestures as rapture, even these can be understood as something evoked through the presence of the Holy Spirit and/or encounters with the *potential* and *praesentia* of the saint.

In the second part of the article, I place the theory of liminality under critique. As Caroline Walker Bynum has pointed out and the case of Gerald of Wales's story so clearly shows, a male gaze on events like that of St. Eluned's feast day tends to focus on certain aspects, and neglect others. The liminality terminology makes sense only from a well-educated, high-status male point of view. In the story of St. Eluned, neither she nor the worshipping community are part of this elite, male framework. So, I ask, can the feast day celebrations be understood in another way? My conclusion is that there are enough strands of evidence in the historical records, hagiographic materials, and descriptions of both the dancing and the materials surrounding the feast to argue that an alternative interpretation is worth pursuing. Not only was Eluned herself a woman who was left without a voice in Gerald of Wales's story, neither are the women following in her footsteps, or the lay community gathering at her feast day, given a focal position in his account. By moving away from the context of liminality and church leadership that describes a ritual of redemption and sin, I present materials that open up the possibility that the local community had developed their own liturgical enactments. In such a liturgy, the dancing in itself formulated a theology for the feast. We may never know exactly what such a feast looked like or the meaning it carried for the participants. I imagine, though, based on the knowledge Caroline Walker Bynum and Jane Cartwright have brought forth about the religious lives of medieval women and the particular female saints in Wales and their following, that the liturgy carried vital elements of thanksgiving and gratitude. Instead of the disruptive and transformational language of a liminal time and/or ritual, the dancing may have been a clearly celebratory and worshipful element in the liturgy. Dancing may have been—even in its more ecstatic and rapturous moments—a pattern for communing with God and the saint's power. I imagine movements and gestures that led the worshipper into deeper levels of intimacy with Christ, rather than afflictions that the priest absolves. I do not claim that Gerald of Wales's story of the feast day of St. Eluned is better understood within or without the conceptual framework of liminality. I merely want to show how different the accounts may become, depending on the point of departure of the interpreter of the story. In both cases, the movements of dance and gesture created theologically significant patterns that still have much to offer researchers wanting to engage with both theology and the arts.

In the end, I indicate that St. Eluned and the telling of the story of her feast day may have carried significance to many generations of people in the community of Brecon, and I look forward to finding more traces of dancing in Wales.

**Funding:** This research was made possible due to yearly salary granted by Runar Eriksson Fond for Theological Students from the Åland Islands, Inez and Julius Polin Institute for Theological Research and Stiftelsen Åbo Akademi (no numbers available).

**Institutional Review Board Statement:** Not applicable.

**Informed Consent Statement:** Not applicable.

**Data Availability Statement:** All images are from my private collection or wikimedia commons.

**Conflicts of Interest:** The author declares no conflict of interest.

## Notes

1   (Coakley 2013; Lloyd 2011, pp. 110, 119).

2   (Caciola 1996, p. 249).

3   (Caciola 1996, p. 249; Mews 2009, pp. 543–44).

4   Gougaud is found in Rahner (1967, pp. 79–80). For the influence of Gougaud on many generations of scholarship, see Hellsten (2021, pp. 62–68, 128, 133, 142–57).

5   (Gougaud 1914, pp. 229–30).

6   (Renberg and Phillis 2021).

7   (Rohmann 2009, pp. 13–45; 2013, 2015, pp. 48–70; Caciola 1996, 2016, pp. 249, 252).

8   (Bynum 1991; Brown 1981).

9   (Chaganti 2018, pp. 4–7; Straple-Sovers 2021, p. 22).

10   (Nevile 2004, p. 6; Nevile 2008; Sjöstrand 2011, pp. 201–4, 219–24).

11   (Lee Eden 2000, p. 214).

12   (Dickason 2020, p. 6).

13   (Coakley 2013, p. 347).

14   (Coakley 2013, pp. 44–51).

15   (Coakley 2013, p. 191).

16   (Coakley 2013, pp. 48–51, 79, 84–85, 267).

17   (Felski 2015, p. 18).

18   The approach is further developed in (Hellsten 2021, pp. 21–44).

19   A great example of this is how Kélina Gotman explains that unruly forms of dancing have been connected to hysteria and other forms of medical explanations (Gotman 2018).

20   He speaks about *reverentia*, which is defined in the following way: "*Reverentia* implied a willingness to focus belief on precise invisible persons, on Christ and his friends the saints—the *amici dominici*—in such a way as to commit the believer to definite rhythms in his life (such as the observation of the Holy days of the saints), to direct his attention to specific sites and objects (the shrines and relics of the saints), to react to illness and to danger by dependence on these invisible persons, and to remain constantly aware, in the play of human action around him that good and bad fortune was directly related to good or bad relations with these invisible persons" (Brown 1981, p. 119).

21   The feasts of the saints circled around the idea of relics. Brown explains that in the cult of relics, "late-antique and early-medieval piety lived down with gusto to his strictures. This cult gloried in particularity. *Hic locus est:* 'Here is the place', or simply *hic*, is a refrain that runs through the inscriptions on the early martyrs' shrines of North Africa. The holy was available in one place, and in each such place it was accessible to one group in a manner in which it could not be accessible to anyone situated elsewhere" (Brown 1981, p. 86). And through this presence—*praesentia*—came the need to regulate who had access, and when, to the power—*potentia*—of the saint. Brown explains that when reading the life story or text of the liturgy where the saint was described—the *passio*—the past is brought into the present. Reading the *passio* activates the relic. "Coinciding as it did with the high point of the saint's festival, the reading of the *passio* gave a vivid, momentary face to the invisible *praesentia* of the saint. When the *passio* was read, the saint was 'really' there: a sweet scent filled the basilica, the blind, the crippled, and the possessed began to shout that they now felt his power in healing, and those who had offended him in the past had good reason to tremble" (Brown 1981, p. 82). Brown also describes that, "In the healing of the possessed, the *praesentia* of the saint was held to be registered with unfailing accuracy, and their ideal power, their *potentia*, shown most fully and in the most reassuring manner. For the solemn and dramatic course of possession and exorcism formed an exact fit with expectations of the exercise of 'clean' power which ( . . . ) clustered around the tombs of the saints. Late-antique and early-medieval men were not merely impressed by the melodramatic associations of exorcism: they felt that in such a drama they witnessed more clearly and with greater precision the manner in which God, through his lords the saints, could stretch forth into their midst the right hand of his healing power" (Brown 1981, p. 109).

22    Coakley's *Théologie totale* is planned to include visual arts, music, poetry and liturgy but no specific section on dance. So far, however, only the book featuring mainly paintings has been published. For more on this, see (Coakley 2013, p. 91; Hellsten 2021, p. 43 fn.163).

23    (Straple-Sovers 2021, pp. 22–31).

24    (Straple-Sovers 2021, p. 28).

25    (Straple-Sovers 2021, p. 27).

26    (Straple-Sovers 2021; Betts 2017, pp. 157–61; Gilchrist 2012).

27    (*The Routledge Handbook of Sensory Archaeology* 2020).

28    (Van Gennep 2013).

29    (Rohmann 2009, pp. 29, 34; 2013, 2015, pp. 53–65).

30    (Turner 1969).

31    For the later development, see (Turner and Abrahams 2017; Thomassen 2009).

32    (Dickason 2020, p. 70).

33    (Thomassen 2014; Björkman 2020).

34    (Brown 1981, pp. 42–43).

35    (Brown 1981, p. 42).

36    (Brown 1981, pp. 42–43).

37    Brown (1981, pp. 37–38, 88–91, 94–95, 105–15). See the methodological sub-chapter, note 17, for explanations of the terms.

38    (Rohmann 2009, pp. 29, 34; 2013, 2015, pp. 53–65; Caciola 1996, 2016, pp. 249, 252).

39    The idea that the Church and Christianity have always been against dancing is commonly found in many other writings as well (Arcangeli 1994; Bashir 2022; LaMothe 2018; Lepeigneux 2022). More nuanced accounts are also emerging (Knäble 2016; Dickason 2020; Knäble 2022).

40    (Rohmann 2009, p. 34; 2015, pp. 53–65; Caciola 1996, 2016, pp. 249, 252).

41    Bynum (2007, pp. 7–8, 52–55, 64, 71–75). As already stated in the introduction, Walker Bynum does not write about dancing per se, she writes about practices that would have been unsettling.

42    (Brown 1981, pp. 82–83, 88–101; Le Goff 2014, pp. 56–67, 150–151, 158–161; Bynum 2007, pp. 77–80; Caciola 1996).

43    For more on examples of this kind, see Hellsten (2021, pp. 233–35, 244–52).

44    Similar ideas, however, not explicitly stated in this manner, are brought forth by Laura Clark in her description of the Vitae of St. Edith of Wilton, where the cursed carolers make an appearance (Clark 2021, pp. 109–17, 120–23).

45    "Gerald of Wales, or Giraldus Cambrensis, was born at Manorbier in Pembrokshire in around 1146. His real name was Gerald de Barri, and he was of mixed Welsh and Norman ancestry. His father, William de Barry, was a leading Welsh nobleman. His uncle was Bishop of St. Davids and he received a religious education. He became chaplain to King Henry II of England in 1184. He accompanied Prince John on his expedition to Ireland in 1184, which led to his first book, *Topographia Hibernica* (1188). In 1188 he accompanied the Archbishop of Canterbury, Baldwin of Exeter, on a tour of Wales recruiting for the Third Crusade, which led to him writing the *Itinerarium Cambriae* (1191) and the *Descriptio Cambriae* (1194). He died in about 1223" (Wales 2009).

46    Caciola (1996, p. 41; 2016, p. 249). Today, the whole of *Itinerarium Cambriae* is found online as an illustrated site with available maps and images (Wales 2009). However, the text used in this article comes from *Giraldi Cambrensis Opera* (2010).

47    (Watkins 2007, p. 101).

48    (Heng 2018, pp. 38–39).

49    (*Giraldi Cambrensis Opera—Gemma Ecclesiastica* 2012).

50    (Watkins 2007, p. 101).

51    "CAP. XLIII. Quod saltationibus et cantilenis in ecclesiis et cemeteriis populi vacare non debent" (*Giraldi Cambrensis Opera—Gemma Ecclesiastica* 2012, p. 119).

52    "Irreligiosa consuetudo est quam vulgus per sanctorum solemnitates agere consuevit. Populi qui debent divina officia attendere saltationibus turpibus invigilant, cantica non solum mala canentes, sed et religiosorum officiis obstrepunt. Hoc etenim ut ab omnibus provinciis depellat sacerdotum ac judicum a concilio sancto curse committatur" (*Giraldi Cambrensis Opera—Gemma Ecclesiastica* 2012, pp. 119–20).

53    "Nemo in oratorio aliquid agat nisi ad quod factum est, unde et nomen habet" (*Giraldi Cambrensis Opera—Gemma Ecclesiastica* 2012, p. 120).

54    (*Giraldi Cambrensis Opera—Itinerarium Kambriae et Descriptio Kambriae* 2012).

55    This date is derived from the translation of "eodem in loco, singulis annis in capite kalendarum Augusti" (*Giraldi Cambrensis Opera—Itinerarium Kambriae et Descriptio Kambriae* 2012, p. 32). I want to thank Joonas Vanhala for the help provided with translating the Latin texts.

56 (Butler 1981, p. 239).

57 The site of the chapel is thought to be near Slwch Tump just outside Brecon, according to (Morgan 1903), cited by (Watkins 2007, p. 100).

58 (Driver 2014).

59 There is a specific journey one can take in the footsteps of St. Eluned; however, the website about it has expired. This indicates that the tradition of St. Eluned has not completely vanished, yet is still not fully alive (Eluned's Way & Wellsites 1999). This is in contrast to other medieval pilgrimage practices containing elements of dance, which are told to have been in continuous use (Hellsten 2021, pp. 146–48).

60 "Erat autem antiquitus regionis illius, quse Brecheniauc dicitur, dominator vir potens et nobilis, cui nomen Brechanus; a quo et terra Brecheniauc denominata. De quo mihi notabile videtur, quod ipsum viginti quatuor habuisse filias historic Britannicse testantur, omnes a pueritia divinis deditas obsequiis, et in sanctitatis assumptee proposito vitam feliciter terminasse. Extant autem basilicae adhuc per Kambriam multae, earum nominibus illustratae: quarum una in provincia de Brecheniauc, non procul a castro principali de Aberhotheni, in collis cujusdam vertice sita, qate Sanctae Aelivedhae ecclesia dicitur: hoc etenim virginis sanctae nomen extiterat, quae et ibidem terreni regis nuptias respuens, aeterno nubens regi, felici martyrio triumphavit" (*Giraldi Cambrensis Opera—Itinerarium Kambriae et Descriptio Kambriae* 2012, pp. 31–32). This translation is mainly built on the text in (Wales 2009).

61 The medieval idea of sanctified space being created through the presence and interaction of a Christian ruler, churches and holy people can be found in writings by a variety of authors (Taylor 2007, pp. 24–30, 39, 44–46, 446–47; Duby 1981; Terpstra 2015; Le Goff 2014; Zika 1988).

62 (Butler 1981, p. 239).

63 (Brown 1988).

64 (Bynum 1991, p. 202).

65 This theme will be further developed in the second part of this article when I return to both the female point of view of this story and the particularities of the Welsh situation.

66 (Taylor 2007, pp. 71, 86, 107–108, 439–440; Brown 1981, pp. 75–77).

67 (Butler 1981, pp. 239–40).

68 "Celebratur autem solemnis ejusdem dies, eodem in loco, singulis annis in capite kalendarum Augusti: ubi et eodem die multi de plebe longinquis ex partibus convenire solent; et variis languentes infirmitatibus, meritis beatae virginis, optatam recipere sanitatem consueverant. Mud autem hoc in loco mihi notabile videtur, quod in omni fere solemnitate hujus virginis accidere consuevit. Videas enim hie homines seu puellas, nunc in ecclesia, nunc in coemiterio, nunc in chorea qua circa coemiterium cum cantilena circumfertur, subito in terram corruere, et primo tanquam in extasim ductos et quietos, deinde statim tanquam in phrenesim raptos exsilientes, opera quaecunque festis diebus illicite perpetrare consueverant, tarn manibus quam pedibus coram populo repraesentantes" (*Giraldi Cambrensis Opera—Itinerarium Kambriae et Descriptio Kambriae* 2012, p. 32). This translation is mine, done with help from Joonas Vanhala and the English version found in (Wales 2009).

69 (Hellsten 2021, pp. 29–32, 40–50, 55–57; Coakley 2013, pp. 44–51, 71–92; Knibbe and Kupari 2020).

70 (Brown 1981, pp. 1–20, 119–26).

71 Examples of this can be found in (Arcangeli 1994, 2017; Ulvros 2004; LaMothe 2018), and the contrary in rare examples like (Silen 2008, pp. 67–78; Knäble 2022).

72 Contrary to this, Watkins argues that it is very likely that Gerald of Wales could have one tone in his more legal texts addressing church practices (*Gemma Ecclesiastica*), and then turn to a more pastoral role when giving the depictions in *Itinerarium Cambriae* (Watkins 2007, pp. 102–3). In her article on dancing in the early Church, Camille Lepeigneux does a really good job of contextualising the negative comments on dance but leaves out all of the positive ones, which distorts the general image (Lepeigneux 2022).

73 The arguments around what constituted sanctioned and sacred dances can be found in (Tronca 2019; Dickason 2020).

74 (Caciola 2006; Caciola and Sluhovsky 2012; Caciola 1994; Katajala-Peltomaa and Toivo 2020).

75 (Katajala-Peltomaa 2020, pp. 419–31).

76 (Brown 1981, pp. 106–27).

77 Liturgical depictions from the medieval period exemplify a procession by describing clear rows of seven ranks of people separating clergy from laity and men from women and children (De Voragine 1993, pp. 278–80). This idealised depiction further carries symbolic meaning as the worship on earth resembles the ranks of angels continuously worshipping in heaven (Hellsten 2021, pp. 182–87). Dickason exemplifies the more common understanding of mixed gender interaction with references to both John Chrysostom (d. 407) and medieval priests that wanted to prevent dancing due to its capacity to bring men and women together. She writes: "The intermingling of men and women during holy days was particularly alarming. It unleashed temptation, thereby distracting priests and devolving saints' feasts into bacchanals" (Dickason 2020, p. 60).

78 Caciola (2016, pp. 249–50). Unfortunately, there is not space in this article to look further into the particular meanings of the space of cemetery for this account. The connection between dancing and cemeteries is explored further here: Adrien Belgrano, "Danses profanes et lieux sacrés au Moyen Âge central: Les Danses dans les cimetières entre contrôle social et négocations" (Belgrano 2016). Both Caciola and Schmitt combine the story of St. Eluned with scholarship focusing on ghosts, cemeteries and dancing (Schmitt 1998). However, theologically speaking encounters with ghosts and celebrations of martyrs or saints are not similar kinds of events. As Caciola writes, saints are special kinds of dead people (Caciola 1996, pp. 36–39).

79 Brown (1981, pp. 42–49). Emphasis on the importance of the liturgical period of the Christmas tide is, for example, completely forgotten in the most recent scholarship around the Kölbick dancers in (Renberg and Phillis 2021). For examples of the contrary, see (Harris 2011; Dickason 2020, pp. 236–24).

80 "Exemplum de sacerdote, qui in Anglia Wigornise finibus Ms nostris diebus interjectam quandam cantilense particulam, ad quam ssepius redire consueverant, quam refectoriam seu refractoriam vocant, ex reliquiis cogitationum, et quoniam ex abundantia cordis os loqui solet, quia tota id nocte in choreis circiter ecclesiam ductis audierat, mane ad niissam sacerdotalibus indutus, et ad aram stans insignitus, pro salutatione ad populum, scilicet, 'Dominus vobiscum' eandem Anglica lingua coram omnibus alta voce modulando pronuntiavit in hunc modum: "Swete" lamman dhin are'. Oujus hsec dicti mens esse potest: 'Dulcis arnica, tuam poscit amator opem'. Hujus autem eventus occasione episcopus loci illius, Willelmus scilicet de Norhale, sub anathematis interminatione publice per synodos et capitula prohiberi fecit, ne cantilena ilia, propter memorise refricationem, quse ad mentem facinus revocare posset, de csetero per episcopatum suum caneretur" (*Giraldi Cambrensis Opera—Gemma Ecclesiastica* 2012, p. 120).

81 (Watkins 2007, p. 102).

82 (Watkins 2007, p. 100).

83 (Encyclopedia Britannica: Cantilena 1998 Accessed on 15 August 2022; Dickason 2020, pp. 133, 203, 265).

84 Similar arguments have been made for dance descriptions in (Tronca 2019).

85 Angelico (1425, 1435). Image rights wikimedia commons.

86 Ms. 23 (86.ML.606), fol. 90v. See https://www.getty.edu/art/collection/object/107SXY#full-artwork-details (accessed on 1 May 2022)

87 f. 108v–109r.

88 (Beumkes 2020 (Accessed on: 15 August 2022); Hellsten 2021, pp. 296–97, 313–22).

89 (Smith 1999).

90 (*Giraldi Cambrensis Opera—Itinerarium Kambriae et Descriptio Kambriae* 2012, p. 32).

91 See (Wales 2009; Gougaud 1914; Backman 1945, pp. 225–32). Trance is also the translation used in (Watkins 2007, p. 100). However, his reading is more nuanced.

92 For suggestions of further reading see (Bourguignon 2004).

93 The complete sentence is (*Giraldi Cambrensis Opera—Itinerarium Kambriae et Descriptio Kambriae* 2012, p. 32).

94 For dance and ecstasy in the Spirit see (Tronca 2016; Dickason 2020, pp. 142–44, 166–68).

95 (Coakley 2013, pp. 116–35, 294–95, 312–22; Harrison 2013, p. 225).

96 "Non ergo illa deliciarum comes, atque luxuriae praedicatur saltatio, sed qua unusquisque corpus attollat impigrum, nec humi pigra jacere membra vel tardis sinat torpere vesligiis. Saltabat spiritaliter Paulus, cum se pro nobis extenderet, et posteriora obliviscens, priora appetens, contenderet ad bravium Christi. Tu quoque cum ad ba- ptisimum venis, manus elevare, pedes quibus ad aetema conscendas, velociores habere moneris (Phil. Ii, 13, 14). Haec saltatio fidei socia, gratia comes". Ambrose of Milan (n.d., CSEL, vol. 73, pp. 181–82), see (Schaff 1890, CCEL, 616).

97 (Coakley 2013, pp. 116–35, 294–95, 312–22; Heiding 2022, pp. 17–18).

98 These are aspects of a worship service prevalent in many Black churches even today (Holmes 2017).

99 "Deinde statim tanquam in phrenesim raptos exsilientes" (*Giraldi Cambrensis Opera—Itinerarium Kambriae et Descriptio Kambriae* 2012, p. 32).

100 (Dickason 2020, p. 168).

101 1 Samuel 10:5–6,10–11,13; 1 Samuel 19:20; 1 Chronicles 25:1,2,3.

102 David's dance in front of the Ark can be found in: 2 Samuel 6:1–7, 14–15; 1 Samuel 10:6b–7 and 1 Chronicles 13, 15. Saul dancing in a frenzy under the prophets can be found in: 1 Samuel 10, 19:18–24. More on this (Witherington 1999).

103 2 Corinthians 12:2–4.

104 (Shantz 2009).

105 John 16:13.

106 I state that Gerald of Wales interprets the passage in this way because, in this section, he not only describes what he sees, he goes further into naming that why the people are doing these gestures is due to some "work they have unlawfully done on feast days". However, this is just his interpretation. As the second part of this article shows, the dancers themselves may have experienced that their dancing stemmed from something completely different.

107 "Videas hunc aratro manus aptare, ilium quase stimulo boves excitare; et utrumquasequasi laborem mitigando, solitas barbarae modulationis voces efferre. Videas hunc artem sutoriam, illium pellipariam imitari. Item vidquaseanc, quasi colum bajulando, nunc filum manibus et brachiis in longum extrahere, nunc extractum occando tanquam in fusum revocare; istam deambulando productis filis quasi telam ordiri; illam sedendo quasi jam orditam oppositis lanceolae jactibus, et alternis calamistrae cominus ictibus, texere mireris. Demum vero intra ecclesiam cum oblationibus ad altare perductos, tanquam experrectos et ad se redeuntes obstupescas" (*Giraldi Cambrensis Opera—Itinerarium Kambriae et Descriptio Kambriae* 2012, pp. 32–33). The translation into English is directly from (Wales 2009).

108 This same habit is still (in Sir Richard Colt Hoare's time) "used by the Welsh ploughboys; they have a sort of chaunt, consisting of half or even quarter notes, which is sung to the oxen at plough: the countrymen vulgarly supposing that the beasts are consoled to work more regularly and patiently by such a lullaby" (Wales 2009, fn.44).

109 I want to thank Joonas Vanhala in particular for the help with translations and understanding of the Latin in this section.

110 (Watkins 2007, pp. 99–101).

111 (Watkins 2007, p. 101).

112 "Demum vero intra ecclesiam cum oblationibus ad altare perductos, tanquam experrectos et ad se redeuntes obstupescas" (*Giraldi Cambrensis Opera—Itinerarium Kambriae et Descriptio Kambriae* 2012, p. 33).

113 "Do your best to present yourself to God as one approved by him, a worker who has no need to be ashamed, rightly explaining the word of truth". 2 Timothy 2:15; "And you who were once estranged and hostile in mind, doing evil deeds, he has now reconciled in his fleshly body through death, so as to present you holy and blameless and irreproachable before him—provided that you continue securely established and steadfast in the faith, without shifting from the hope promised by the gospel that you heard, which has been proclaimed to every creature under heaven. I, Paul, became a servant of this gospel". Colossians 1:21–23; "I appeal to you therefore, brothers and sisters, by the mercies of God, to present your bodies as a living sacrifice, holy and acceptable to God, which is your spiritual worship. Do not be conformed to this world, but be transformed by the renewing of your minds, so that you may discern what is the will of God—what is good and acceptable and perfect". Rom. 12:1–2.

114 Interestingly, Ambrose's text, where dancing in the Spirit is mentioned (Ambrose of Milan n.d., *De Poenitentia.* 73, pp. 181–82), is concerned with penitence, and he also speaks about the need to press on and present ourselves before God.

115 (Brown 1981, pp. 35–38, 89–100).

116 (Watkins 2007, p. 101).

117 His translation in (Watkins 2007, p. 101).

118 (Watkins 2007, p. 101).

119 For more on the birth of purgatory, see (Le Goff 1984, pp. 4–5; Schmitt 1998, pp. 4–5; Caciola 2016, p. 151).

120 "Ubi et eodem die multi de plebe longinquis ex partibus convenire solent; et variis languentes infirmitatibus, meritis beatae virginis, optatam recipere sanitatem consueverant" (*Giraldi Cambrensis Opera—Itinerarium Kambriae et Descriptio Kambriae* 2012, p. 32).

121 (Brown 1981, p. 100).

122 (Brown 1981, p. 100).

123 Walker Bynum writes that when looking at medieval accounts one cannot even presume that sickness and disease were always linked to sin, death and the devil. Sometimes illness was even perceived as a gift that drew people closer and more intimately towards God (Bynum 1991, pp. 189–91).

124 (Watkins 2007, pp. 105–6).

125 (Lloyd 2011, p. 110).

126 (Lloyd 2011, p. 119).

127 Rohmann (2015, pp. 54, 61). In his account the status of dancing in the views of the Christian churches is much more ambivalent than what I express here.

128 (Lloyd 2011, pp. 68–69, 94–96, 130).

129 (Watkins 2007, p. 106).

130 (Bynum 1991, pp. 27–43).

131 (Bynum 1991, pp. 32–34).

132 (Bynum 1991, pp. 34–40).

133 (Coakley 2013, pp. 84–85).

134 "De quo mihi notabile videtur, quod ipsum viginti quatuor habuisse filias historic Britannicse testantur, omnes a pueritia divinis deditas obsequiis, et in sanctitatis assumptee proposito vitam feliciter terminasse" (*Giraldi Cambrensis Opera—Itinerarium Kambriae et Descriptio Kambriae* 2012, p. 31).

135 (Bynum 1991, p. 48).

136 (Bynum 1991, pp. 48, 149–50, 172–73, 198).

137 (McAvoy 2010, pp. 211–12; Cartwright 2008, pp. 123–24).

138 (McAvoy and Hughes-Edwards 2005, pp. 8–14; Mulder-Bakker 2005, pp. 1–3).

139 McAvoy and Hughes-Edwards (2005, pp. 11–13). At the same time, the authors also show how many distortions have been created particularly around the phenomena of female recluses due to an eroticising and mysticising tendency found in earlier, predominantly male authorship, on the topic (Mulder-Bakker 2005; McAvoy and Hughes-Edwards 2005, pp. 6–9, 11–14).

140 (Hayward 2005; Mulder-Bakker 2005, p. 3).

141 (Cartwright 2008, pp. 123–24).

142 (McAvoy and Hughes-Edwards 2005, pp. 7–18).

143 (McAvoy and Hughes-Edwards 2005, p. 13).

144 Santha Bhattacharji further points out that a specific form of a feminised concept of enclosure is developed with the story of the hermit Guthlac (673–714). To withdraw into the tomb and the solitary confinement it offers is not only a break with marriage, betrothals and social obligations. It was also perceived as stepping into the womb of the virgin space. This can be described as a step away from others and a step into an intensified life of prayer, desire, and union (McAvoy and Hughes-Edwards 2005, p. 13).

145 (McAvoy and Hughes-Edwards 2005, pp. 7–18; McAvoy 2010).

146 Both Brown's accounts from the Early Church and Nancy Caciola's descriptions from the medieval period show how female leadership and authority at times were something that even threatened the Church (Brown 1988; Caciola 2016).

147 Cathedral Church of St John Evangelist, Brecon, Powys visited 24 September 2019.

148 (Lloyd 2011, p. 130).

149 (Lloyd 2011, p. 131).

150 (Lloyd 2011, p. 131).

151 (Cartwright 2008, pp. 70–71, 92).

152 (Bynum 1984, pp. 247–62; 1991, pp. 34–38, 154; Weinstein and Bell 1982, pp. 123–37, 228–32).

153 (Cartwright 2008, pp. 70–83).

154 (Cartwright 2008, p. 85).

155 (Cartwright 2008, pp. 85–86).

156 This pattern is seen even when the authors of the texts describing these women are male (Cartwright 2008, pp. 85–86).

157 This is also the reason I opted for not reading too much into the connections found between rapture and rape, in the first part of this article. Such choices run the risk of accentuating a male gaze and inscribing more drama into the women's lives in a way that renders them merely passive objects of suffering and affliction.

158 (Cartwright 2008, p. 86).

159 (Cartwright 2008, pp. 76–77).

160 (Mooney n.d.).

161 (McAvoy 2010, p. 212; Cartwright 2008, p. 94).

162 (Lloyd 2011, p. 110).

163 (Lloyd 2011, pp. 130–31).

164 (Lloyd 2011, p. 119).

165 (Lloyd 2011, p. 110).

166 (Worcestre 1969, p. 155; Cartwright 2008, p. 71).

167 (Cartwright 2008, pp. 202–8).

168 (Cartwright 2008, p. 71).

169 (Brown 1981, pp. 42–45, 111).

170 (Brown 1981, pp. 98–101; Terpstra 2015).

171 The account of Gerald of Wales actually continues, after giving his version of the feast of Eluned, with a section that describes the riches of the land (fertility) and how blessed the region is (*Giraldi Cambrensis Opera—Itinerarium Kambriae et Descriptio Kambriae 2012*, p. 33).

172 (Brown 1981, pp. 79–82).

173 (Cartwright 2008, pp. 72–75).

174 (Felski 2015, p. 18; Chaganti 2018, pp. 14, 46, 122).

175 Thus, the rest of this article paints the feast of St. Eluned in broader strokes, still aiming to highlight what can be gained from leaving the male gaze on the celebration in the margins.

176 (Cartwright 2008, p. 4).

177   (Brown 1981, pp. 79–82).

178   (Cartwright 2008, pp. 71–72).

179   (Cartwright 2008, p. 72).

180   For the importance of blood in veneration of saints, see (Bynum 2007).

181   (Cartwright 2008, pp. 72–75).

182   We can read about Hagar who named God "the one who sees" in Genesis 16, the Samarian woman who Jesus met and who
      has been seen as one of the first missionaries in John 4, Paul and Timothy who encountered Lydia—a woman who became an
      important figure in the local Christian community—in Acts 16, and the prophetess Miriam from the Exodus story.

183   (Cartwright 2008, pp. 202–8).

184   (Cartwright 2008, pp. 175–77).

185   The window was commissioned in 1910 and installed in that year or in 1911. It was given in memory of Philip Howel Morgan
      (1816–1868) of Defynnog, rector of Llanhamlach; his wife Margaret (1816–1884) of Buckingham Place, Brecon; and Margaret's
      brother William Hughes (1820–1886), vicar of Ebbw Vale (Crampin 2019).

186   (People's Collection Wales 2009).

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
