# Peer review of "The Liminal Space of Medieval Dance Practices: The Case of St. Eluned’s Feast Day"

_arts, 2022_

Round 1

Reviewer 1 Report

I very much enjoyed reading “The Liminal Space of Medieval Dance Practices: The Case of St. Eluned’s Feast Day,” and I recommend it for publication, with minor revisions.

This article submission presents an intriguing study of dance, Western medieval Christianity, and critical theory. The article is well-researched, and incorporates canonical thinkers (Victor Turner, Peter Brown, etc.) alongside scholars who are not ask well known in Anglophone contexts (Gregor Rohmann). I appreciated the inclusion of works by Nancy Caciola, who is a fine medievalist and should be cited more in current scholarship than she currently is. It was exciting to see Gerald of Wales used in a new way, as his text will be familiar to most medievalists but not so much his incorporation of dance. The author draws upon Bynum’s important critique of Victor Turner’s liminality, which, as Bynum shows, does not always fit the medieval evidence, or at least not in the formulaic way that Turner put forth.

This article is already very strong, but I have a few suggestions for improvement. The author mentions trance and gives several biblical examples. But I found the concept of trance under-theorized, especially from an anthropological perspective. Perhaps Erika Bourguignon may be useful: Erika Bourguignon, “Suffering and Healing, Subordination and Power:/Women and Possession Trance,” Ethos 32:4 (2004; Bourguignon, Trance Dance, repr. in Dance Perspectives 35 (Autumn 1968); Bourguignon, Possession (Prospect Heights:/Waveland Press, 1991).

I also felt that more context on cemetery dances would help. Consider these references: Adrien Belgrano, “Danses profanes et lieux sacrés au Moyen Âge central: Les Danses dans les cimetières entre contrôle social et négociations,” European Drama and Performance Studies 8 (2016):-25– 42; Jean-Claude Schmitt, Ghosts in the Middle Ages; Caciola, Afterlives.

On the issue of ghosting in dance, Seeta Chaganti’s Strange Footing book may also be useful.

Author Response

Thank you very much for this reading and review!

On the suggestions of improvement I will mostly add comments in the footnotes. 

This is because the discussion on trance and possession - albeit important for the story of Gerald of Wales - is a very complex and complicated one. I think it deserves much more space, as it will also look very different depending on the perspective the analysis derives from. I thank you for Erika Bourguignon’s article - this will be very helpful for a new approach to this same story.

Furthermore, the comments on cemetery dances will, unfortunately (as the other reviewer requested many other alternations) need to be kept for an article that focuses on the landscape of this event. Not only the cemetery but also the holy spring and the placement of the shrine outside the city centre, are all important aspects that need further discussion.

Finally, Seeta Chaganti’s Strange Footing will need to be mentioned - so this I attend to in the new section on methods.

Reviewer 2 Report

This article addresses a fascinating text and early medieval textual example of dancing and asks interesting and original questions about it. However, many of the questions are unfortunately left unanswered or are answered with relatively weak arguments or interpretations. Some sections of the article are stronger than others. I would very much like to see the article in print and believe it could be a valuable addition to scholarship on medieval depictions of dance, particularly textual descriptions, but the piece needs extensive revision and careful editing to reach that point.

General comments:

-Terms need to be defined and context needs to be more thoroughly provided throughout. This article would be read by a non-specialist audience (in this case, a non-medievalist audience) that will probably not be familiar with terms such as potentia and praesentia, a hermeneutics of suspicion and charity, and kinesics. You are also engaging here with several concepts that are presented with the assumption of audience understanding--for example, on page 7 you describe a long row of mixed-gender and -aged people dancing together and say "such breaking of traditional social roles" as if everyone knows which social roles you mean, and you mention the "activation of the relic" several times without explaining what that is. Be cognizant of the fact that non-medievalists will most likely read this piece.

-Citations need to be thoroughly checked--you often use the wrong format or cite things incorrectly (Caroline Bynum Walker rather than Walker Bynum, citing the whole edited collection rather than the actual chapter you are quoting in a footnote and in the bibliography).

-The piece should also be carefully copyedited--the standard of English varies and there are many examples of sentences that are a bit unclear, awkward, or grammatically incorrect.

-There are some suggestions you make that need to be elaborated upon or more thoroughly supported: for example, fn 20, 21, 25 (what are you quoting? Citation is missing), your point that virginity for women functioned like chastity for men on p. 6, etc.

-I’m not clear what purpose the images are serving, and you need to provide figure captions for them.

-Several times, you bring in sources from other times and places and even point out that they are indeed from other times and places. Why are you bringing them in, then? Are they relevant to this text or your interpretation of it? Your arguments for bringing them in are not currently strong enough.

Introduction: Other than some awkward phrasing, this section is fine.

Dance and Theories of Liminality:

-"Liminality" is spelled wrong in the heading.

-In general I found this section convincing although some elements could be clarified for a non-specialist audience.

-Your use of Peter Brown’s quotation on pilgrimage is effective as another example of active entry into a liminal state, like dance.

-You need to define potentia and praesentia.

-You position your own work well near the end of this section, in the paragraph beginning “Very few dance scholars or medieval historians…”

-Define “activation of the relic.”

Gerald of Wales and the Itinerarium Cambriae:

-You refer to “Gougaud” without introducing that author; maybe there was an earlier reference that has been deleted?

-You consistently misspell Kölbigk as Krölbigk

-The first two sentences of the second paragraph should be simplified—right now this piece is very long, wordy, and convoluted.

-I think you presume too much in the first paragraph of p. 5 (“One could presume that if G of W found problematic aspects of lived Christian faith…” This either needs to be supported better or perhaps stated differently to be less of an unsupported suggestion.

-You give one quotation from GoW about Church Fathers condemning dance. I think you need to strengthen this point as a foundation for your following argument—that it is significant, then, that Gerald does not seem to condemn the dancers at the Eluned feast.

-Your first block quote on page 5 is not currently indented, so it doesn’t appear to be a quotation

-In some footnotes, you give the Latin version of the text, and in others, you don’t. You should consistently give the original language quotation.

-“Latin” and “English” should always be capitalized.

-You give English translations from “Wales, ‘The Itinerary…’” but you don’t give the translator’s name.

-Your point that virginity functioned for women like chastity did for men on page 6 needs contextualizing. CWB makes this argument in her book, but does she make it about this time and place? Is that argument relevant in early/mid medieval Wales? For non-specialists, the difference between chastity and virginity and their gendered significance may not be clear. This may need to be explained, and this is a very broad statement with only one supporting citation—I would recommend providing other sources on this topic.

-Why is this picture here? And what is the source?

-You need to give a more thorough explanation of a hermeneutics of suspicion and charity. Where is this term from? Who else has done this work that you’re building on?

-I would avoid saying “Gerald of Wales did not find the disorderly dancing problematic,” as you can’t actually know that. Your argument here is that the dancing is not represented or depicted as problematic in the text. (p. 7)

-You say he didn’t find it problematic even though it is “outside the ‘norm’ of the anniversary of a Saint.’” (Saint does not need to be capitalized here). According to whom? This is an unsupported assertion.

-You do not give a background or explanation of what a kinesic reading is. This is important and needs to be clarified.

-On p. 8 you mention “the kind [of dancing] that Gerald condemned in his other writings.” Where? What kind does he condemn? This is not made clear in your analysis. You mention one quotation from a Church Father earlier in the essay, but that is not enough of a background or foundation to establish.

-P. 8: “Something abruptly happens.” It would be more effective to actually quote the excerpt you’re describing.

-As the article goes on, I think your arguments become more vague. P. 10: “Gerald of Wales interprets this as a moment where earlier hidden aspects of people’s lives are revealed.” Where do we see this? This argument is not elaborated upon and I’m not sure where we see Gerald doing this.

-Several places where you should probably have footnotes: p. 11 after “sacred time and sin,” “approved the ritual as ‘appropriate,” p. 12 after “both monks and scholars”

-Your arguments about purgatory at the top of page 12 need more support.

-Paragraph 2 (after the block quote) on page 12 needs to be clarified. I’m not following your line of argument.

End of the article:

Rather than continuing the list of suggestions, I would summarize my comments on the final sections here:

I give many comments above because I think those sections are very worth revising and find them interesting and engaging. They could definitely be improved and I encourage you to do so! I’m less clear about the last two sections of the article and I find myself a bit lost about your arguments here. I’m unclear as to how these arguments relate to your reading of Eluned, especially the kinesic details. You seem to abandon the kinesic reading here. You also seem to have some significant logic problems and inconsistencies here—you say that being a bride of Christ is a continuation of a woman’s normal life and doesn’t “turn [a woman’s] life upside down,” but how would fleeing the safety of her home and leaving the worldly life of marriage and childrearing not do that? I can sort of see the argument you are gesturing at here, but there are some serious problems in its sequence and how you make it. There is a lot of speculation in these last two sections, and a lot of interesting questions asked, but you leave us with almost all of those questions unanswered rather than using them to introduce arguments.

Regarding your point that many Welsh saints were not canonized and thus we don’t have official vitae, etc.—most early medieval English saints were also not canonized, and we have plenty of vitae, homilies, liturgies, etc., about them.

The end of the article takes a distinctly personal, reflective, and speculative turn. The beginning is grounded in research and textual analysis, and I encourage you to keep that up throughout the article and to stay grounded in your kinesic reading (which can be developed and strengthened).

Author Response

There are many comments to all the suggestions in the file. Thank you so much for helping me in making this article so much better!

Round 2

Reviewer 2 Report

This is definitely an improved article in terms of overall clarity, soundness of argument, and methodology. The added sections do a lot to improve it and the second half seems much stronger and better framed. The conclusion is a wonderful addition that lessens the feeling of vagueness and the sense of many unanswered questions.

I would still recommend heavy editing of the language and citation style. There are many awkward sentences throughout and grammatical problems (you consistently neglect the possessive when talking about Gerald of Wales's writing, for example). There are also multiple typos or misspellings throughout.

Also, the sources are not accurately cited. When citing a chapter in an edited book, for example, you must cite the actual chapter name and the page range of the chapter, not just provide the author's name and then the general book information. And that chapter must get its own entry in the references list--right now, for example, you have an entry for the Miller and Phillis book but not one for Straple-Sovers as the actual chapter you're citing.

Author Response

This is now a version that has gone through revision of the references and professional language check.

Thank you so much for the review work!